# Whole-cortex in situ sequencing reveals input-dependent area identity

Xiaoyin Chen[1,7 ✉], Stephan Fischer[2,7], Mara C. P. Rue[1,7], Aixin Zhang[1], Didhiti Mukherjee[3], Patrick O. Kanold[3,4], Jesse Gillis[5 ✉] & Anthony M. Zador[6 ✉]

The cerebral cortex is composed of neuronal types with diverse gene expression that are organized into specialized cortical areas. These areas, each with characteristic cytoarchitecture[1,2], connectivity[3,4] and neuronal activity[5,6], are wired into modular networks[3,4,7]. However, it remains unclear whether these spatial organizations are reflected in neuronal transcriptomic signatures and how such signatures are established in development. Here we used BARseq, a high-throughput in situ sequencing technique, to interrogate the expression of 104 cell-type marker genes in 10.3 million cells, including 4,194,658 cortical neurons over nine mouse forebrain hemispheres, at cellular resolution. De novo clustering of gene expression in single neurons revealed transcriptomic types consistent with previous single-cell RNA sequencing studies[8,9]. The composition of transcriptomic types is highly predictive of cortical area identity. Moreover, areas with similar compositions of transcriptomic types, which we defined as cortical modules, overlap with areas that are highly connected, suggesting that the same modular organization is reflected in both transcriptomic signatures and connectivity. To explore how the transcriptomic profiles of cortical neurons depend on development, we assessed cell-type distributions after neonatal binocular enucleation. Notably, binocular enucleation caused the shifting of the cell-type compositional profiles of visual areas towards neighbouring cortical areas within the same module, suggesting that peripheral inputs sharpen the distinct transcriptomic identities of areas within cortical modules. Enabled by the high throughput, low cost and reproducibility of BARseq, our study provides a proof of principle for the use of large-scale in situ sequencing to both reveal brain-wide molecular architecture and understand its development.

The vertebrate brain is organized into subregions that are specialized in function and distinct in cytoarchitecture and connectivity. This spatial specialization of function and structure is established by developmental processes involving intrinsic genetic programs and/or external signalling[10]. Although gene expression can change during cell maturation and remains dynamic in response to internal cellular conditions and external stimuli, a core transcriptional program that maintains cellular identity usually remains steady in mature neurons[11]. Thus, resolving the expression of core sets of genes that distinguish different types of neuron provides insight into the functional and structural specialization of neurons.

Many large brain structures are spatially organized into divisions, or modules, within which neurons are more similar in morphology, connectivity and activity. In the cortex these modules usually involve a set of adjacent cortical areas that are highly interconnected[3,4,7] and correlated in neuronal activity[5,6]. Many cortical areas also share the same medium- and fine-grained transcriptomically defined neuronal types[9,12]. Whether and how the areal and modular organization of cortical connectivity and activity is reflected in the transcriptomic signatures of areas is unknown.

To address this question, here we apply BARseq[13,14] to interrogate gene expression and the distribution of excitatory neuron types across nine mouse forebrain hemispheres at high spatial resolution. BARseq is a form of in situ sequencing[15] in which Illumina sequencing-by-synthesis chemistry is used to achieve a robust readout of both endogenous messenger RNAs and synthetic RNA barcodes. These RNA barcodes are used to infer long-range projections of neurons. We have previously used BARseq to identify the projections of neuronal types defined by gene expression[14,16] and/or their locations[13,17], and to identify genes associated with differences in projections within neuronal populations[14]. Importantly, we showed that BARseq can resolve transcriptomically defined cell types of cortical neurons at cellular resolution by sequencing dozens of cell-type markers[14]. Because BARseq has high throughput and low cost compared with many other spatial techniques[18–24], it is ideally suited for studying the spatial organization of gene expression at cellular resolution over whole-brain structures such as the cortex.

[1]Allen Institute for Brain Science, Seattle, WA, USA. [2]Institut Pasteur, Université Paris Cité, Bioinformatics and Biostatistics Hub, Paris, France. [3]Department of Biomedical Engineering, Johns Hopkins University, Baltimore, MD, USA. [4]Kavli Neuroscience Discovery Institute, Johns Hopkins University, Baltimore, MD, USA. [5]Department of Physiology, University of Toronto, Toronto, Ontario, Canada. [6]Cold Spring Harbor Laboratory, Cold Spring Harbor, NY, USA. [7]These authors contributed equally: Xiaoyin Chen, Stephan Fischer, Mara C. P. Rue. ✉e-mail: xiaoyin.chen@alleninstitute.org; jesse.gillis@utoronto.ca; zador@cshl.edu

Here we use BARseq as a standalone technique for sequencing gene expression in situ at brain-wide scale in nine animals, with or without binocular enucleation, to resolve the distribution of neuronal populations and gene expression across the cortex. We generate high-resolution maps of 10.3 million cells with detailed gene expression, including 4,194,658 cortical cells. We find that, although most neuronal populations are found in multiple cortical areas, the composition of neuronal populations is distinct across areas. The neuronal compositions of highly connected areas are more similar, suggesting a modular transcriptomic organization of the cortex that matches cortical hierarchy and modules defined by connectivity in previous studies[3,4,7]. By comparing littermates with and without binocular enucleation, we then show that peripheral inputs have a critical role in shaping cortical gene expression and area-specific cell-type compositional profiles.

## BARseq maps brain-wide gene expression

Recent single-cell transcriptomic studies[8,12,25–27] have used different nomenclatures to refer to cell types across hierarchical levels. To avoid confusion we first define our cell-type nomenclature. The highest hierarchical level, or H1 type, divides neurons into excitatory neurons, inhibitory neurons and other cells; this level is the 'class' level in many studies. Within each H1 type we subdivide neurons into H2 types, which are sometimes referred to as 'subclasses'[8,9]. Cortical excitatory neurons fall into nine H2 types that are shared across most cortical areas. This division refines the traditional projection-based intratelencephalic (IT)/pyramidal tract (PT)/corticothalamic (CT) neuron classification[28] as follows: PT and CT neurons correspond to L5 extratelencephalic neurons (ET) and L6 CT neurons, respectively, whereas IT neurons are subdivided into L2/3 IT, L4/5 IT, L5 IT, L6 IT, NP (near-projecting neurons), Car3 and L6b. This division follows recent single-cell RNA sequencing studies but differs from the classical tripartite classification of IT/PT/CT neurons. Each H2 type can be further divided into H3 types ('cluster' or 'type' level in some studies[8,9]). Previous reports showed that H1 and H2 types are largely shared across most cortical areas, but the expression of many genes is localized to specific parts of the cortex both during development[10,29] and in the adult[30]. Clusters at the H3 level appear to be enriched in neurons from different parts of the cortex[8,12,31], but the detailed distribution of neuronal populations at this higher granularity across cortical areas remains unclear.

To assess the distribution of neuronal populations across the cortex we first generated a pilot dataset by applying BARseq to interrogate the expression of 104 cell-type marker genes (Supplementary Table 1) in 40 hemibrain coronal sections covering the whole forebrain in one animal (Fig. 1a,b). We applied the same approach that we used previously to resolve excitatory neuron types in the motor cortex[14] (Supplementary Note 1, Fig. 1c and Extended Data Fig. 1a–f show marker gene selection and overall strategy), and found 2,167,762 cells across the whole hemisphere. Removal of cells with an insufficient number of reads (20 reads per cell and five genes per cell minimum) resulted in 1,259,256 cells after quality control (Supplementary Methods), with a mean of 60 unique reads per cell and 27 genes per cell (Extended Data Fig. 1g,h). At the gross anatomical level many genes were differentially expressed across major brain structures and cortical layers (Fig. 1a). These expression patterns were consistent with in situ hybridization patterns in the Allen Brain Atlas[30] (Extended Data Fig. 1i and Supplementary Note 1). Thus, our pilot dataset recapitulated the known spatial distribution of gene expression.

## BARseq distinguishes neuronal types

We next identified transcriptomic types of neurons by de novo and hierarchical clustering based on single-cell gene expression in the pilot dataset (Fig. 2a and Supplementary Methods). Clustering all cells resulted in 24 clusters, which we then combined into three H1 types (642,340 excitatory neurons, 427,939 inhibitory neurons and 188,977 other cells) based on the expression of *Slc17a7* and *Gad1* (Extended Data Fig. 2a and Supplementary Methods). Of these 1.2 million cells, 517,428 were in the cortex and were the focus of our analyses. Based on the fraction of excitatory neurons expressing both *Slc17a7* and *Gad1*, we estimated that the probability of segmentation errors in which two neighbouring cells were merged (that is, doublet rate) would be 5–7% (Extended Data Fig. 2b,c and Supplementary Note 2). The 24 clusters, comprising the three H1 types, largely corresponded to coarse anatomical structures in the brain (Fig. 2b). For example, different clusters were enriched in the lateral and ventral groups of the thalamus, the intralaminar nuclei, the epithalamus, the medial, basolateral and lateral nuclei of the amygdala, the striatum and the globus pallidus (Fig. 2b). These results recapitulate the clear distinction of transcriptomic types across anatomically defined brain structures as observed in whole-brain, scRNA-seq studies[26,31–34].

We then reclustered the excitatory and inhibitory neurons separately into H2 types (Fig. 2a,c and Extended Data Fig. 2d) to improve the resolution of clustering. At this level we recovered major inhibitory neuron subclasses (Pvalb, Sst, Vip/Sncg, Meis2-like and Lamp5), all excitatory subclasses that are shared across the cortex (L2/3 IT, L4/5 IT, L5 IT, L6 IT, L5 ET, L6 CT, NP, Car3 and L6b) and an excitatory subclass specific to the medial cortex (RSP) observed in previous cortical scRNA-seq datasets[8,9,12,35]. The H2 types expressed known cell-type markers and other highly differentially expressed genes (Fig. 2d). For example, *Cux2* is expressed mostly in superficial-layer IT and Car3 neurons, *Fezf2* in NP and L5 ET neurons and *Foxp2* specifically in L6 CT neurons (Supplementary Note 3 provides a detailed description). Although we generated the full 40-section dataset in two batches (Supplementary Methods) we did not observe strong batch effects, as evidenced by the intermingling of excitatory neurons from different slices across the two batches in the uniform manifold approximation and projection (UMAP) plot (Extended Data Fig. 2e). Thus the H2 types recapitulated, at medium granularity, known neuronal types identified in previous scRNA-seq datasets[8,9,12].

We then reclustered each excitatory H2 type into H3 types (Fig. 2a). To quantify how well H3 types corresponded to reference transcriptomic types identified in previous scRNA-seq studies, we used a *k*-nearest-neighbour-based approach to match each H3 type to leaf-level clusters recorded in ref. 9 (Supplementary Methods). We found that cortical H2 types had a one-to-one correspondence with subclass-level cell types in the scRNA-seq data (Fig. 2e). Within each H2 type, the H3 types differentially mapped onto single or small subsets of leaf-level clusters in the scRNA-seq data (Fig. 2e; Extended Data Fig. 2f shows matching of clusters outside of the cortex). Both H2 and H3 types were organized in an orderly fashion along the depth of the cortex, recapitulating the laminar organization of cortical excitatory neurons (Extended Data Fig. 2g,h and Supplementary Note 3). At a coarse spatial resolution the H3 types were also found in cortical areas similar to matching clusters in previous scRNA-seq datasets (Extended Data Fig. 2i–k and Extended Data Fig. 3). For example, the H3-type PT AUD and its corresponding scRNA-seq cluster (242_L5_PT CTX) were both enriched in lateral cortical areas (TEa-PERI-ECT) and auditory cortex (AUD), whereas H3-type PT CTX P and its corresponding scRNA-seq clusters (245_L5_PT CTX and 259_L5_PT CTX) were enriched in the visual cortex. Therefore, these results demonstrate that our pilot dataset resolved fine-grained transcriptomic types of cortical excitatory neurons that were consistent with previous scRNA-seq datasets[9] and recapitulated their areal and laminar distribution[9,12,36]. The high resolution and cortex-wide span of our dataset now enabled us to resolve the spatial enrichment of gene expression and the distribution of neuronal subpopulations across the cortex at micrometre-level resolution.

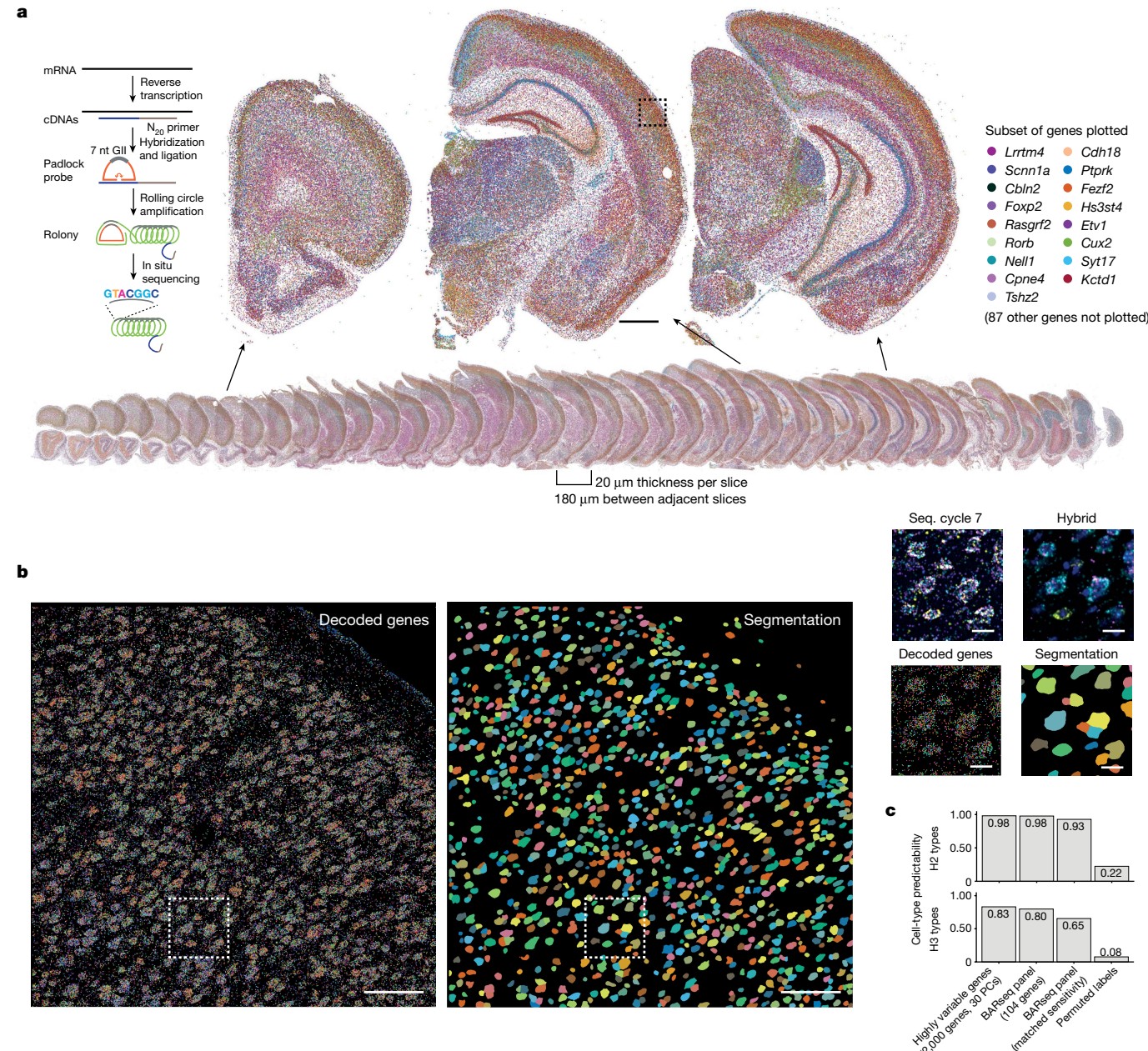

**Fig. 1 | BARseq reveals brain-wide gene expression. a**, Images showing mRNA reads of all 40 slices (bottom) and close-up images of three representative slices (top). For clarity, only 17 out of 104 genes (indicated on the right) are plotted. Inset on the left shows an illustration of mRNA detection using BARseq. **b**, Left, decoded genes and cell segmentations (middle) from a representative imaging tile (out of 4,385 tiles across 40 slices) corresponding to the dashed box in **a**. Right, close-up images of this area showing the last sequencing cycle, hybridization cycle, decoded genes and cell segmentation. **c**, Single-cell cluster assignment performance using the full transcriptome, top principal components (PCs) and the 104-gene panel with or without subsampling to match the sensitivity of BARseq for H2 (top) and H3 (bottom) clusters. Scale bars, 1 mm (**a**), 100 μm for full-tile images (**b**), 10 μm for the boxed area (**b**). cDNA, complementary DNA.

## Gene expression patterns across the cortex

Gene expression varies substantially across the cortex[30,37] but most cortical areas largely share the same H2 types, or subclasses, of excitatory neurons[9,12]. Therefore it is unclear how differences in the organization of neuronal subpopulations lead to area-specific gene expression. Three sources of variation could contribute to gene expression differences across areas (Fig. 3a). First, the composition of H2 types may drive differences in gene expression across the cortex (Fig. 3a (left), the cell-type composition model). For example, the ratio of H2-type X to -type Y might be high in the visual but low in the motor cortex, so genes that are expressed more highly in X than in Y will be more highly expressed in the visual cortex. Second, the expression of some genes may vary across space regardless of H2 type—that is, they change consistently across space in multiple H2 types (Fig. 3a (middle), the spatial gradient model). In this model, gene A may be more highly expressed in the visual than in the motor cortex in types X and Y. Finally, the expression of some genes may vary across space in an H2-specific manner (Fig. 3a (right), the area-specialized cell-type model). For example, gene A may be more highly expressed in the visual cortex than in the motor cortex in H2 type X but not in H2 type Y.

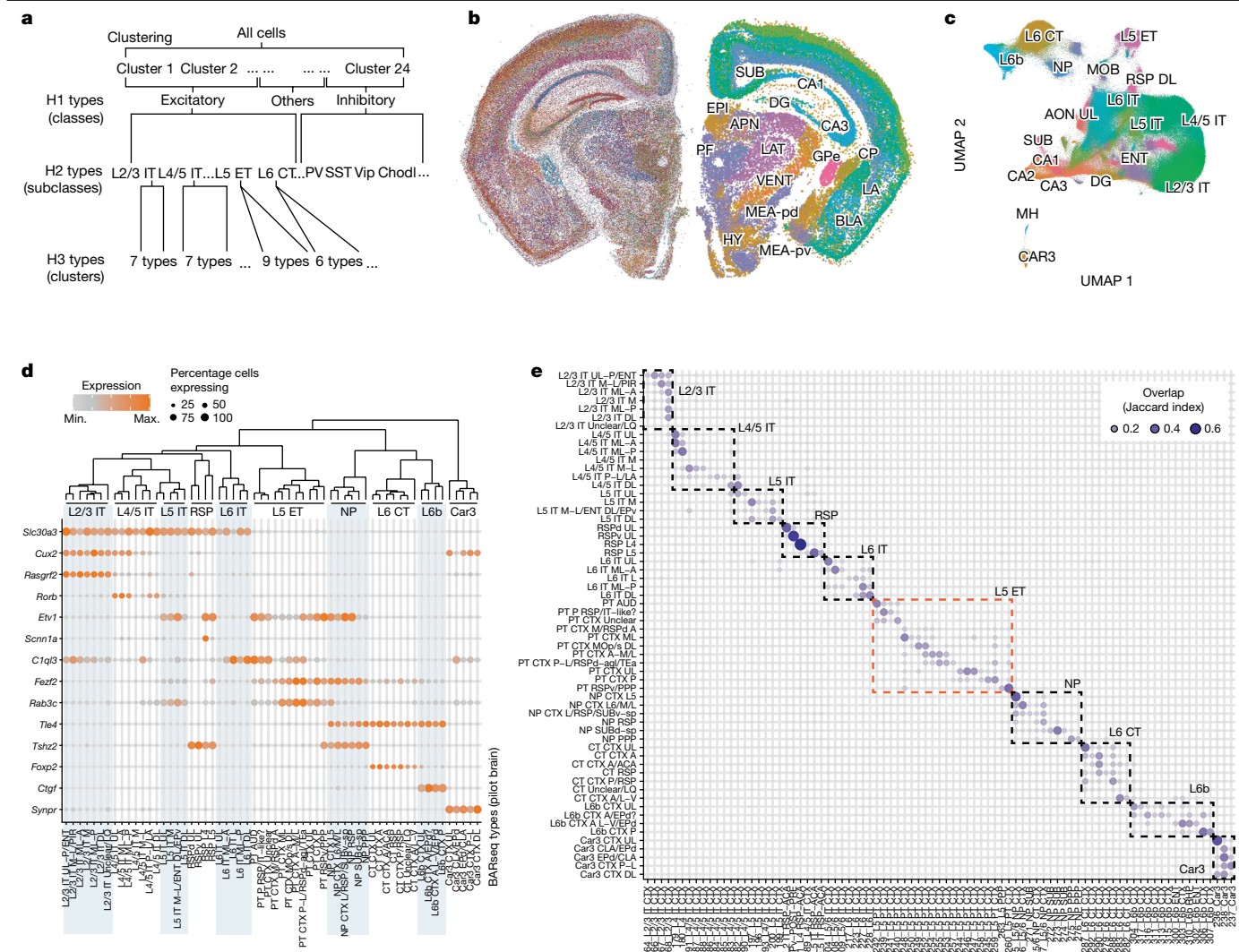

**Fig. 2 | BARseq captures gene expression and spatial distribution of cortical excitatory cell types. a**, Workflow of hierarchical clustering. **b**, Gene expression (left) and H1 clusters (right) in a representative slice. Major anatomical divisions distinguished by H1 clusters are labelled: SUB, subiculum; DG, dentate gyrus; CP, caudate putamen; GPe, globus pallidus, external segment; LA, lateral amygdala; BLA, basolateral amygdala; MEA-pd (pv), medial amygdalar nucleus, posterodorsal (posteroventral); EPI, epithalamus; APN, anterior pretectal nucleus; LAT, lateral group of the dorsal thalamus; VENT, ventral group of the dorsal thalamus; PF, parafascicular nucleus; HY, hypothalamus. **c**, UMAP plot of

gene expression of excitatory neurons, coloured by H2 type. CA, cornu Ammonis; MOB, main olfactory bulb; MH, medial habenula; AON, anterior olfactory nucleus; RSP, retrosplenial area. **d**, Marker gene expression in cortical excitatory H3 types. Colours indicate mean expression level and dot size indicates fraction of cells expressing the gene. The dendrogram (top) shows hierarchical clustering of pooled gene expression within each H3 type. **e**, Overlap (Jaccard index) between BARseq H3 types and scRNA cell types[9]. Dashed boxes indicate parent H2 types. Min., minimum; max., maximum.

To determine the contribution of each source to the variation in gene expression across areas we discretized the cortex on each coronal slice into 20 spatial bins (Supplementary Methods and Extended Data Fig. 4a). We then assessed how much of the variation in bulk gene expression across bins could be explained by either space or composition of H2 or H3 types using one-way analysis of variance (Extended Data Fig. 4b,c and Supplementary Methods). We found that all three models contribute to the spatial variation of gene expression, and that the model that contributes most to variation varies across genes (Extended Data Fig. 4b–d and Supplementary Note 4). Because the spatial patterns of many genes were similar, we sought to extract basic spatial components that were shared across genes and H2 types using non-negative matrix factorization (NMF)[38] (Supplementary Note 4 and Extended Data Fig. 4e,f). We found that the majority of NMF components were

patterned not in broad gradients along major spatial axes, but rather were concentrated in areas that were functionally related and highly interconnected (Fig. 3b and Extended Data Fig. 4g). For example, NMF5 was found mostly in visual areas whereas NMF8 was predominantly in somatosensory areas. Other NMF modules, including NMF1 (medial areas) and NMF10 (lateral areas), were present in combinations of areas that were functionally distinct but also highly interconnected[3,4]. Spatially variant genes were usually strongly associated with only one or two components (Fig. 3c and Extended Data Figs. 4h and 5), and the association recapitulated known spatial patterns of these genes. For example, *Tenm3* was expressed mostly in posterior sensory areas including the visual cortex, auditory cortex and part of the somatosensory cortex[30] (Extended Data Fig. 4d, bottom); *Tenm3* was strongly associated with NMF5 (Fig. 3c), which was also

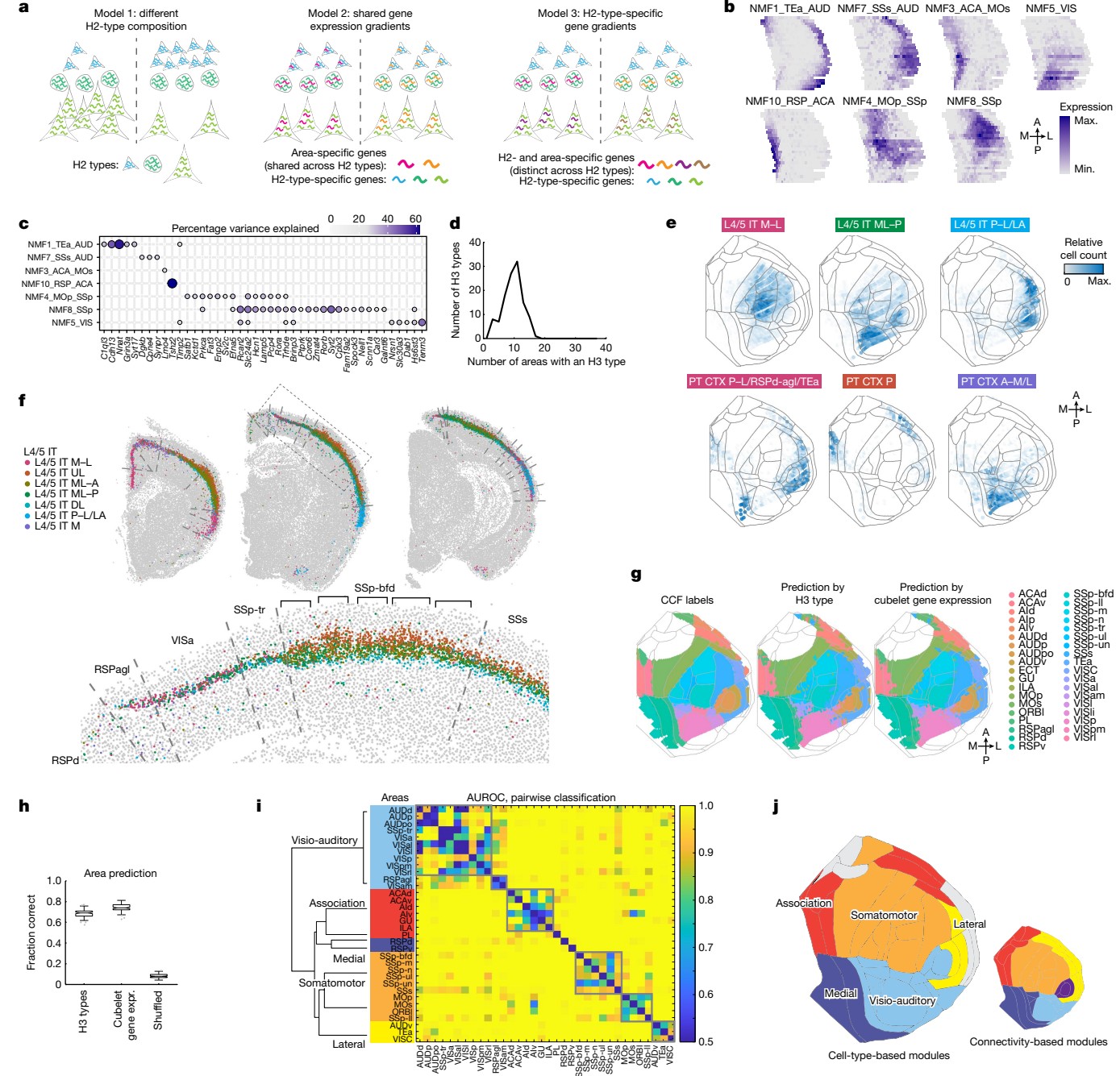

**Fig. 3 | Spatial variations of gene expression across the cortex. a**, Three models of differential gene expression across cortical areas. H2 types are indicated by cell shape; cool-toned shapes indicate H2-type-specific genes; warm-toned shapes indicate area-specific genes in model 2 and H3-type-specific genes in model 3. **b**, Expression of selected spatially variant NMF modules plotted on cortical flatmaps. **c**, Expression of selected marker genes in each NMF module. **d**, Histogram showing the number of areas (out of 37) in which an H3 type was detected: an H3 type was considered present in an area if that area contained at least 3% of that H3 type. **e**, Spatial distribution of example L4/5 IT and L5 ET H3 types across the cortex plotted on cortical flatmaps. Colours indicate relative cell count in each cubelet; grey lines delineate cortical areas. **f**, Distribution of L4/5 IT H3 types in an example coronal section. Dashed lines indicate area borders in CCF. Magnified views of dashed boxes are shown on the

right. Brackets indicate barrels in the barrel cortex. **g**, Cortical areas defined in CCF (left) and those predicted by H3 type (middle) and cubelet gene expression (right). **h**, Fraction of correctly predicted cubelets using H3-type composition, cubelet gene expression and shuffled control. Each box shows the performance of *n* = 100 resampled trials. Boxes show median, and quartiles and whiskers indicate range after exclusion of outliers. Dots indicate outliers. **i**, Matrix showing the AUROC of pairwise classification between combinations of cortical areas. Areas are sorted by modules, which are colour coded on the left. The dendrogram was calculated using similarity of H3-type composition; clusters were obtained based on the matrix and are shown in the grey-bordered boxes. **j**, Cortical flatmaps coloured by cell-type-based modules (left) and by connectivity-based modules identified by Harris et al.[3] (right).

expressed in the same sets of areas (Fig. 3b). Thus, gene expression varies along sets of interconnected areas, suggesting an intriguing link between gene expression and intracortical connectivity across areas.

## Cell-type-defined cortical modules

The spatially varying NMF modules were obtained after controlling for variability in the composition of H2 types, but not of H3 types. Therefore

we hypothesized that these modules reflected differences in the composition of H3 types across cortical areas. Consistent with this hypothesis, each H3 type was enriched in a small subset of NMF modules and H3 types also overlapped with their corresponding NMF modules in space (Extended Data Figs. 6 and 7a,b, Supplementary Methods and Supplementary Note 5). To further assess the areal distribution of H3 types we rediscretized the cortex on each coronal slice into 'cubelets' of similar width along the mediolateral axis across all slices (Supplementary Methods and Extended Data Fig. 4a). These cubelets were of similar physical size and were narrower on the mediolateral axis than the spatial bins used in the previous analysis; this higher lateral resolution makes it easier to assign cubelets to individual cortical areas. We found that H3 types were shared by multiple cortical areas and were not specific to any single area (each H3 type was found in between six and 12 areas, median ± 1 s.d., Fig. 3d; Fig. 3e shows distributions of example H3 types and Supplementary Fig. 1). Thus the distinctness of neighbouring cortical areas cannot be explained simply by the presence or absence of an area-specific H3 type. However, we noticed that the compositional profiles of H3 types often changed abruptly near area borders defined in the Allen Common Coordinate Framework v.3 (CCF)[39] (Fig. 3f and Extended Data Fig. 7c,d). Most salient changes occurred at the lateral and medial areas, which is consistent with scRNA-seq data[9]. Within the dorsolateral cortex, although neighbouring cortical areas sometimes shared sets of H3 types, their proportions typically changed at or near area borders. Using either gene expression or the compositions of H3 types in each cubelet, we could accurately predict cubelet locations and cortical area labels (75% correct using gene expression and 69% correct using H3-type composition, compared with 8% in shuffled control; Supplementary Note 5, Fig. 3g,h and Extended Data Fig. 7e–i). Thus both cubelet gene expression and H3-type composition are highly predictive of locations along the tangential plane of the cortex and the identity of the cortical areas.

We next assessed the similarity and modularity of cortical areas based on how well these could be distinguished by their H3-type composition (Fig. 3i and Supplementary Methods). In brief, we built a distance matrix between cortical areas based on how well they can be distinguished pairwise using H3 type composition then performed Louvain clustering on the distance matrix. We identified six clusters, each of which consisted of more than one area (Fig. 3i, grey-bordered boxes); these included two clusters corresponding to the visio-auditory areas and one cluster each for the association areas, somatosensory cortex, motor cortex and lateral areas. This modular organization is robust to small errors in CCF registration (Extended Data Fig. 7j and Supplementary Methods). We further combined these clusters with singlet areas (PL, RSPd and RSPv) that did not cluster with any other area into cortical modules based on similarity in H3-type composition. These modules largely included the visio-auditory, somatomotor, association, medial and lateral areas, respectively (Fig. 3i). Notably, these cell-type-based modules were largely consistent with cortical modules that are highly connected (connectivity-based modules)[3] (Fig. 3j). Thus, highly interconnected cortical areas share similar groups of H3 types and, consequently, characteristic transcriptomic signatures.

## Cell types are robust to enucleation

Transcriptomic types, areas and modules reflect cortical organization at different scales, suggesting that they may be generated through different developmental mechanisms. As a first step in understanding the developmental processes that contribute to cortical organization at different scales, we applied BARseq to examine how the postnatal removal of peripheral sensory input alters the organization of cortical transcriptomic types. Thalamocortical projections have a central role in shaping the identities and borders of cortical areas[10,40,41], and loss of postnatal visual inputs affects gene expression in VISp and other areas[40,42,43]. How peripheral inputs shape cortical neuronal types and

the characteristic cell-type compositional profiles of cortical areas, however, is unclear. For example, altered gene expression could result in new cell types that are not seen in a normal brain; alternatively, it could enrich or deplete existing cell types (Fig. 4a). Because BARseq is cost effective and has high throughput, it is uniquely suited for interrogating changes in neuronal gene expression and cell-type compositional profiles on a brain-wide scale across many animals, with or without developmental perturbation.

We performed binocular enucleation on four mice at postnatal day 1 and collected their brains at postnatal day 28, along with those of four matched littermate controls ($n$ = 8 animals) (Fig. 4b). We performed BARseq using an improved microscope that achieved better data quality and much faster data acquisition compared with the pilot dataset (2.3 days per brain; Supplementary Methods and Supplementary Note 6). In total, the full dataset contained 9.1 million quality controlled cells covering most of the forebrain of all eight animals (Fig. 4b), with a median of 87 reads per cell and 37 genes per cell (Fig. 4c). Cells from individual brains were interdigitated with those from other brains in UMAP space, suggesting that there were minimal batch effects (Fig. 4d and Extended Data Fig. 8a). Therefore, we performed de novo clustering hierarchically on the concatenated data of 3,957,252 excitatory neurons, 1,526,182 inhibitory neurons and 3,635,402 other cells at the H1 level. The fraction of other cells was significantly higher than that in the pilot dataset, probably because the improved data quality allowed more cells with lower read counts to pass quality control and be included. We then reclustered the excitatory neurons into 35 H2 types (Fig. 4e) and 154 H3 types, including 12 H2 types and 70 H3 types predominantly found in the cortex. These H3 types in the new dataset closely matched those in the pilot dataset (Extended Data Fig. 8b,c; Supplementary Note 6 shows mapping to the pilot dataset). Notably, no H3 type was strongly enriched or depleted in enucleated brains compared with control (Extended Data Fig. 8f; Supplementary Note 7 and Extended Data Fig. 8d–i provide detailed analyses). Although we cannot fully rule out the possibility that minor changes in gene expression were missed at our transcriptomic resolution, these results suggest that enucleation did not lead to the creation of new cell types at the H3 level; rather, the main effect of enucleation was probably reflected in changes in the compositional profiles of H3 types.

## Enucleation alters cell-type make-up

Having established that enucleation did not create new H3 types, we sought to characterize enucleation-induced changes in area-specific H3-type composition. We divided the cortex into cubelets using an approach similar to that used for the pilot data (Supplementary Methods). This discretization resulted in about 270 neurons per cubelet, with a mean distance of 181 µm between adjacent cubelets in a section. To visualize H3 type composition we plotted UMAP analysis based on the fraction of H3 types in each cubelet (Fig. 5a–c). Consistent with the absence of batch effects seen in single-neuron gene expression (Fig. 4d), cubelets from all eight animals mixed smoothly in most areas (Fig. 5a). Colour coding of cubelets by condition (Fig. 5b), however, revealed an 'island' (left) within which cubelets from the two populations (enucleated versus control) were largely segregated. This island contained mainly cubelets from VISp and other visual areas (Fig. 5b,c, insets). To quantify differences in the compositional profiles of H3 types between control and enucleated brains we trained a classifier to assess how distinct cubelets from each cortical area were between the two conditions (Supplementary Methods). If enucleation consistently altered the compositional profile of H3 types in a cortical area, then we would expect the classifier to predict whether a cubelet was from a control or an enucleated animal based on its H3 type composition above chance level. In most cortical areas the classifier performed at chance level, but VISp cubelets were highly predictive of condition (Fig. 5d; area under the receiver operating characteristic (AUROC)

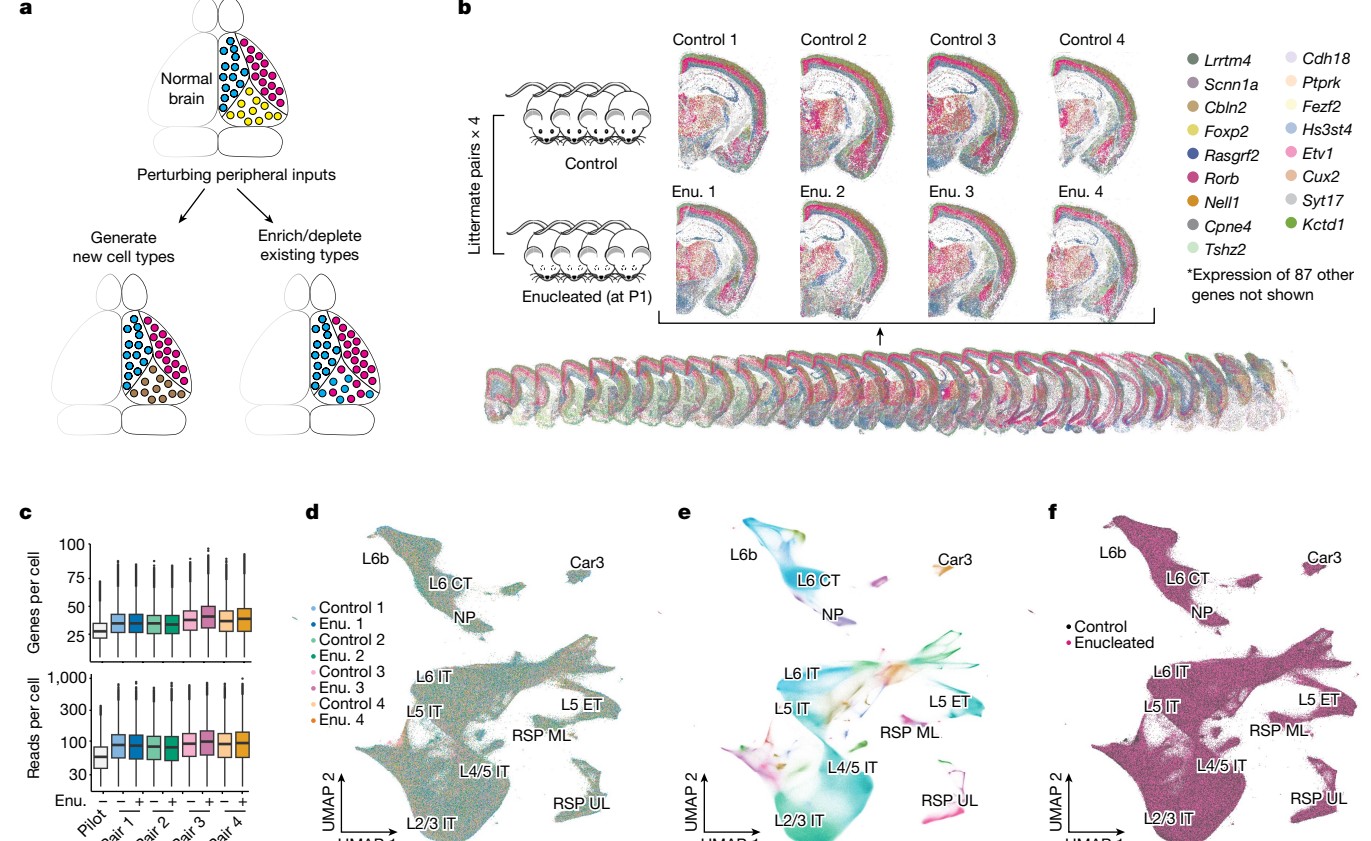

**Fig. 4 | BARseq consistently detects cell types across eight enucleated and control animals. a**, Models of possible effects of removing peripheral sensory inputs postnatally, including generation of new cell types (left) and/or enrichment and depletion of existing cell types (right). **b**, We collected brain-wide transcriptomic data from four littermate pairs: within each pair one mouse was enucleated (Enu.) at P1 and the other was a sham control (left, *n* = 8 animals). A representative stack of 32 slices from one brain (bottom) and close-up images of matching coronal slices from all eight brains (top) are shown. For clarity, only 17 out of 104 genes (indicated on the right) are plotted.

**c**, Genes and read counts per cell for the pilot dataset and the enucleation and control littermates (in order plotted, *n* = 0.6, 1.0, 1.0, 1.1, 1.3, 1.2, 1.1, 1.2 and 1.1 million biologically independent cells). Boxes indicate the median and first and third quartiles; whiskers extend to the most extreme value up to 1.5 times interquartile range from each bound; remaining dots are plotted individually. **d–f**, UMAP plots of gene expression of excitatory neurons from all eight animals. Neurons are colour coded by animal (**d**), by H2 type (**e**) and by condition (enucleated or control, **f**). Labels show only H2 types in cortex.

0.90 ± 0.06 compared with shuffled AUROC 0.56 ± 0.27, median ± s.d.; *P* = 2 × 10⁻³³ using rank sum test and Bonferroni correction). Two higher visual areas (VISpm and VISl; AUROC median ± s.d. 0.70 ± 0.12 and 0.66 ± 0.15; and shuffled AUROC median ± s.d. 0.50 ± 0.25 and 0.44 ± 0.25; *P* = 3 × 10⁻⁹ and 7 × 10⁻⁹, respectively, comparing each area with shuffled control using rank sum test and Bonferroni correction) and a non-visual area (SSp-ll; AUROC 0.57 ± 0.09 and shuffled AUROC 0.42 ± 0.18, median ± s.d.; *P* = 2 × 10⁻⁸ compared with shuffled control using rank sum test and Bonferroni correction) were also predictive above chance level, although the predictive powers were much lower. Thus, enucleation largely affected the relative composition of H3 types within visual areas.

The effect of enucleation can be observed directly in the distribution of H3 types in the primary visual area (Fig. 5e and Extended Data Fig. 9a). For example, many L2/3 IT M–L_2 neurons (Fig. 5e, yellow dots) were found in VISp in control animals, but L2/3 IT L–M neurons (Fig. 5e, green dots) became enriched in VISp in enucleated animals. Similarly, L6 IT DL neurons (Fig. 5e, purple dots) were found in higher numbers in the VISp of enucleated animals compared with control animals. To systematically examine how enucleation affected the compositional profiles of cortical excitatory cell types in each area we looked for H3 types that were enriched or depleted in enucleated brains using an analysis of variance model, adjusting for litter and area effects (Supplementary

Methods). We found that 46 H3 types in 18 areas across the whole cortex were either over- or under-represented in enucleated animals compared with control. VISp had the most H3 types (16) whose compositions were altered by enucleation (Fig. 5f). The affected H3 types were found across most H2 types, with the strongest enrichment or depletion of H3 types of L2/3 IT, L4/5 IT and L6 IT (Fig. 5g). Intriguingly, L6b/CT A–L_2, a transitional type between L6 CT and L6b H2 types usually found only in lateral areas, was also highly enriched in VISp after enucleation. The affected H3 types remained in their characteristic sublaminar positions (Extended Data Fig. 9b) and overall changes were consistent with, but broader than, those observed during dark rearing during the critical period[42] (Extended Data Fig. 9c and Supplementary Note 8). The top enriched H3 types, including L2/3 IT M–L, L2/3 IT L–M, L4/5 IT M–L, L6 IT DL and L6b/CT A–L_2, were all enriched in medial and lateral areas in the control brains, including areas immediately medial and lateral to the visual areas (Extended Data Fig. 9d). Thus, enucleation broadly shifted neurons in VISp towards H3 types that were usually enriched in the medial and lateral areas in control brains.

## Peripheral inputs sculpt area identities

Because enriched H3 types were consistently found in medial and/or lateral areas in control animals, we wondered whether enucleation

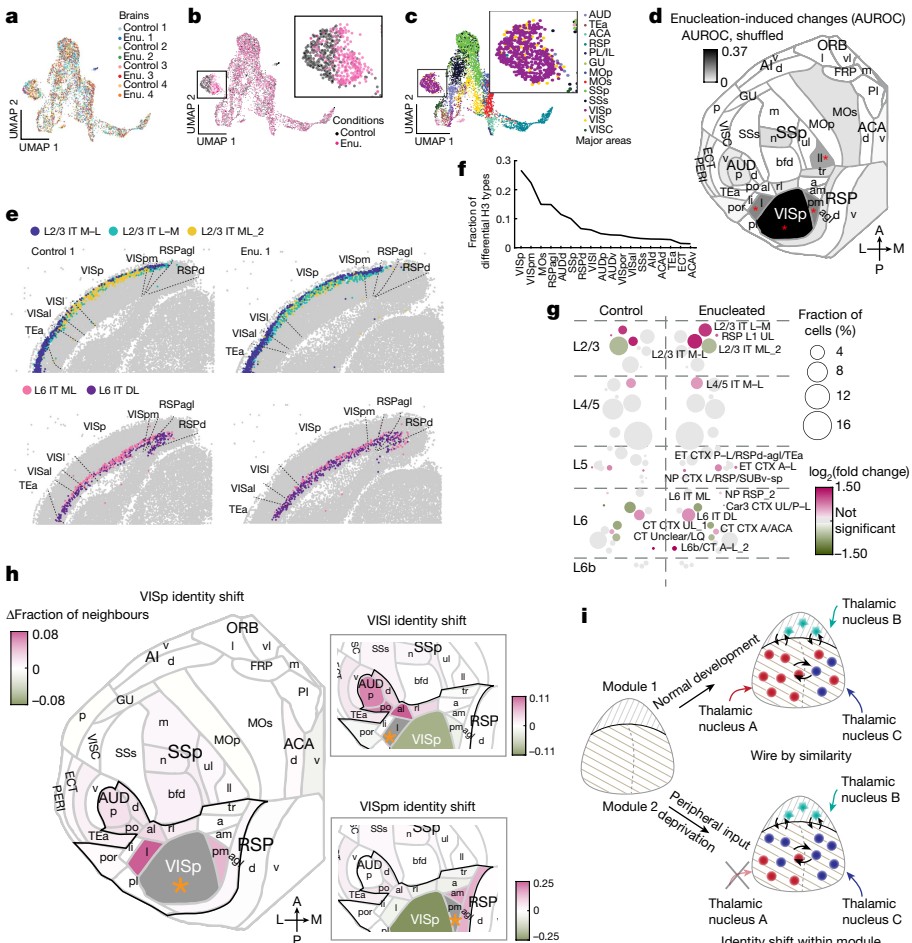

**Fig. 5 | Enucleation alters visual area identities within the visio-auditory module. a–c**, UMAP plots of H3 type composition of cubelets from all eight brains, colour coded by animal (**a**), experimental condition (**b**) and cortical area (**c**). Insets show amplified views of boxed areas. **d**, Flatmap showing how distinct an area was between control and enucleated brains (AUROC). Colours indicate ΔAUROC between scores with or without shuffling of brain conditions. Red asterisks indicate false discovery rate below 0.05. **e**, Representative slices from a pair of littermates showing selected L2/3 IT and L6 IT H3 types enriched or depleted in VISp after enucleation. **f**, Fractions of enriched or depleted H3 types in each area. **g**, Fractions of H3 types in VISp and their fold change after enucleation. Colours indicate log(fold change) and circle size indicates the fraction of VISp neurons belonging to each H3 type. **h**, Shifts in areal identity for VISp (left), VISl (top right) and VISpm (bottom right). Colours indicate

fraction of enriched or depleted neighbours in enucleated brains compared with control. The area of interest in each plot is shaded in grey and indicated by an orange asterisk. Areas outlined in black indicate the visio-auditory module. **i**, Models of cortical cell-type organization. Cortical modules are established independently of peripheral inputs (left). Under normal conditions, cortical areas follow wire-by-similarity (top). Areas within a module have similar cell types and are more interconnected. When peripheral inputs are removed (bottom), the cell-type compositional profile of the affected area shifts towards other areas within the same module. Modules are denoted by patterned backgrounds. Coloured dots and stars indicate cell types in the two modules, respectively; coloured arrows indicate thalamic inputs; black arrows within the cortex indicate connectivity.

had also shifted overall area identity—as defined by the H3-type compositional profiles—of the visual cortex towards other areas. To examine how area identities had changed after enucleation we used a nearest-neighbour-based approach inspired by MetaNeighbor[44] to assess the similarity of cubelets in both control and enucleated brains to other cubelets in control brains (Supplementary Methods). If enucleation had shifted the compositional profile of an area towards a target area, cubelets from the affected area in the enucleated brain would then have had more neighbours in the target area than cubelets from the same area in the control brain. For each cubelet in a littermate pair we found the 20 cubelets with closest match in H3-type composition in control brains from the other three pairs of littermates. We then calculated the similarity, quantified by AUROC for assigning cubelets from each area to areas in the control brains based on nearest neighbours (Extended Data Fig. 9e). All three visual areas (VISp, VISl and VISpm, circled in Extended Data Fig. 9e) remained highly similar to the same areas in control brains (AUROC 0.97 and 0.98 for control and

enucleated VISp, 0.90 and 0.92 for control and enucleated VISl and 0.88 and 0.94 for control and enucleated VISpm, respectively), indicating that their H3-type compositions remained highly distinct from other areas despite the changes induced by enucleation. However, all three visual areas also shifted towards the identities of neighbouring regions as judged by the fraction of neighbours from an area (Fig. 5h). For example, VISp cubelets from the enucleated brains had higher AUROC scores with both VISl and VISpm than those from control brains (0.85 and 0.89 for enucleated cubelets and 0.76 and 0.83 for control cubelets; Extended Data Fig. 9e). Consistent with the high AUROC scores observed, VISp cubelets from enucleated brains also had more neighbours in VISl and VISpm (Fig. 5h). Similarly, VISl cubelets from enucleated brains had more neighbours in auditory areas and VISpm cubelets from enucleated brains had more neighbours in VISam and RSPagl (Fig. 5h, insets). Notably, all three areas shifted towards neighbouring areas that were physically further away from VISp and were within the visio-auditory module (black outlines in Fig. 5h). To examine

whether these changes reflected a shift in area borders or a change in composition across an area, we plotted each cubelet from enucleated brains and coloured them by differences in the number of neighbour cubelets in VISl (Extended Data Fig. 9f, top) and VISpm (Extended Data Fig. 9f, bottom). In VISp the enrichment of neighbours in VISl and VISpm was found in cubelets across the whole area. In particular, cubelets that had more neighbours in VISpm after enucleation (red dots in VISp in Extended Data Fig. 9f, bottom) appeared to be concentrated at the centre of VISp rather than at the borders, suggesting that changes in similarity among these areas reflected an overall change in cell-type composition rather than a shift in area borders. Thus, enucleation shifted the H3-type composition-defined area identities of the visual areas towards neighbouring areas within the visio-auditory module.

## Discussion

Using BARseq, we generated cortex-wide maps of transcriptomic types of excitatory neurons at high transcriptomic and spatial resolution across nine animals. These maps not only elaborate the distribution of cortical excitatory neuron types previously revealed by single-cell studies[9,12], but also provide an 'anchor' to associate other neuronal properties and activity with neuronal types. Thus, our spatial cell-type map provides a foundational resource for understanding the structural and functional specialization of cortical areas. We focused on the cortex, but the same approach can be applied to any other brain region with adequately designed gene panels. When examining large numbers of genes, overcoming optical crowding by computational demixing of overlapping signals[45] and/or optimization of cell segmentation using more recent approaches[24,46,47] may further improve the ability to resolve single-cell gene expression accurately.

Our results suggest that the cell-type compositional profiles of cortical areas reflect their modular organization seen in connectivity studies: cortical areas that are highly interconnected also have similar H3 types (Fig. 5i, top). This 'wire-by-similarity' relationship is not a trivial consequence of cell-type-specific connectivity observed at a cortex-wide scale, because cortical neurons of the same type are not necessarily highly connected (for example, *Sst* neurons[48]). Thus, wire-by-similarity does not describe the connectivity of individual neuronal types but rather reflects how divisions within a large brain region (that is, areas within the cortex) relate to each other in terms of cell types and connectivity. Future studies using BARseq to map the projections of neuronal types at cellular resolution, from multiple cortical areas and at multiple developmental time points, can help resolve the single-cell basis of the wire-by-similarity organization.

The combination of single-cell resolution, high transcriptomic resolution and broad interrogation across many cortical areas allowed us to describe in detail how gene expression and cell-type compositional profiles change after removal of peripheral sensory inputs. Overall, the effects of enucleation suggest that peripheral activity refines the cell-type compositional profiles of cortical areas. Enucleation affected IT neurons in all layers and also L6b/CT neurons, a broader population than the L2/3 IT neurons affected by dark rearing (Fig. 5g)[42]. However, enucleation did not completely abolish the distinction between primary and secondary visual areas, as observed by Chou et al.[40] after genetic ablation of thalamocortical axons (Supplementary Note 8). Thus, together with previous studies, our results suggest a consistent model: the physical connections established by thalamocortical axons are needed to define the primary visual cortex, and peripheral activity sharpens cell-type composition across both the primary visual cortex and neighbouring higher visual areas within a cortical module (Fig. 5i, bottom).

BARseq stands out among spatial transcriptomic methods with its high throughput (about 2.3 days per brain on one microscope), low cost (approximately US $2,000 per brain) and high reproducibility. These features make it possible to compare brain-wide spatial gene expression across many animals, thus providing a path to go beyond a single-reference brain atlas[31,33,34] towards a 'pan-transcriptomic' atlas that captures population diversity. Furthermore, combining interrogation across multiple individuals with perturbations enables the discovery of causal relationships. Whereas we studied the effect of developmental perturbations, the same approach can also be used in neuropsychiatric disease models, ageing studies, cross-species comparison and other experimental perturbations. Our approach based on BARseq can be broadly applied to link brain-wide, network-level dynamics with detailed changes in gene expression in single neurons, and to establish causal relationships between developmental processes and brain-wide cell-type organization.

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

## Reporting summary

Further information on research design is available in the Nature Portfolio Reporting Summary linked to this article.

## Data availability

Raw sequencing images are available from the Brain Image Library (https://api.brainimagelibrary.org/web/view?bildid=ace-dim-pad, https://api.brainimagelibrary.org/web/view?bildid=ace-dim-own, https://api.brainimagelibrary.org/web/view?bildid=ace-dim-owl, https://api.brainimagelibrary.org/web/view?bildid=ace-dim-out, https://api.brainimagelibrary.org/web/view?bildid=ace-dim-orb, https://api.brainimagelibrary.org/web/view?bildid=ace-dim-old, https://api.brainimagelibrary.org/web/view?bildid=ace-dim-off, https://api.brainimagelibrary.org/web/view?bildid=ace-dim-odd and https://api.brainimagelibrary.org/web/view?bildid=ace-cry-zip). Both cell- and rolony-level data are provided at Mendeley Data (https://data.mendeley.com/datasets/8bhhk7c5n9/1 (ref. 49) and https://data.mendeley.com/datasets/5xfzcb4kn8/1 (ref. 50)). Gene panel selection and cell-type assessment were based on data publicly available at https://data.nemoarchive.org/biccn/lab/zeng/transcriptome/[8,9] and https://github.com/shekharlab/mouseVC (ref. 51). Allen CCF v.3 with the 2017 annotation was downloaded from https://community.brain-map.org/t/api-allen-brain-connectivity/2988.

## Code availability

Scripts used for both data processing and data analysis are provided at Mendeley Data (https://data.mendeley.com/datasets/8bhhk7c5n9/1 (ref. 49) and https://data.mendeley.com/datasets/5xfzcb4kn8/1 (ref. 50)) and on GitHub (https://github.com/gillislab/barseq_analysis (ref. 52)).

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

**Acknowledgements** We thank W. Wadolowski for technical support; A. Bhandiwad for support in conversion of CCF to flatmap coordinates; K. Brouner for histology support; and B. Tasic, Z. Yao, B. Long, D.-W. Kim and other members of the Allen Institute for discussions. This work was supported by the National Institutes of Health (nos. 5RO1NS073129, 5RO1DA036913, RF1MH114132 and U01MH109113 to A.M.Z.; R01MH113005 and R01LM012736 to J.G.; U19MH114821 to A.M.Z. and J.G.; 1DP2MH132940 to X.C.; 1R01MH133181 to X.C. and J.G.; and R01DC009607 to P.O.K.), the Brain Research Foundation (no. BRF-SIA-2014-03 to A.M.Z.), IARPA MICrONS (no. D16PC0008 to A.M.Z.) and a Robert Lourie award (to A.M.Z.). A.M.Z. was supported by an Allen Distinguished Investigator Award, a Paul G. Allen Frontiers Group advised grant of the Paul G. Allen Family Foundation. The content is solely the responsibility of the authors and does not necessarily represent the official views of the National Institutes of Health. X.C., M.C.P.R. and A.Z. thank the Allen Institute founder, P. G. Allen, for his vision, encouragement and support.

**Author contributions** X.C. conceived the study. S.F. selected the gene panel. D.M. and P.O.K. designed and performed binocular enucleation. X.C., M.R. and A.Z. collected data. X.C., S.F., M.R., J.G., A.M.Z. and A.Z. analysed data. X.C., M.R., S.F. and A.M.Z. drafted the manuscript. All authors were involved in the interpretation of results and finalization of the manuscript.

**Competing interests** A.M.Z. is a founder and equity owner of Cajal Neuroscience and a member of its scientific advisory board. The remaining authors declare no competing interests.

**Additional information**
**Correspondence and requests for materials** should be addressed to Xiaoyin Chen, Jesse Gillis or Anthony M. Zador.

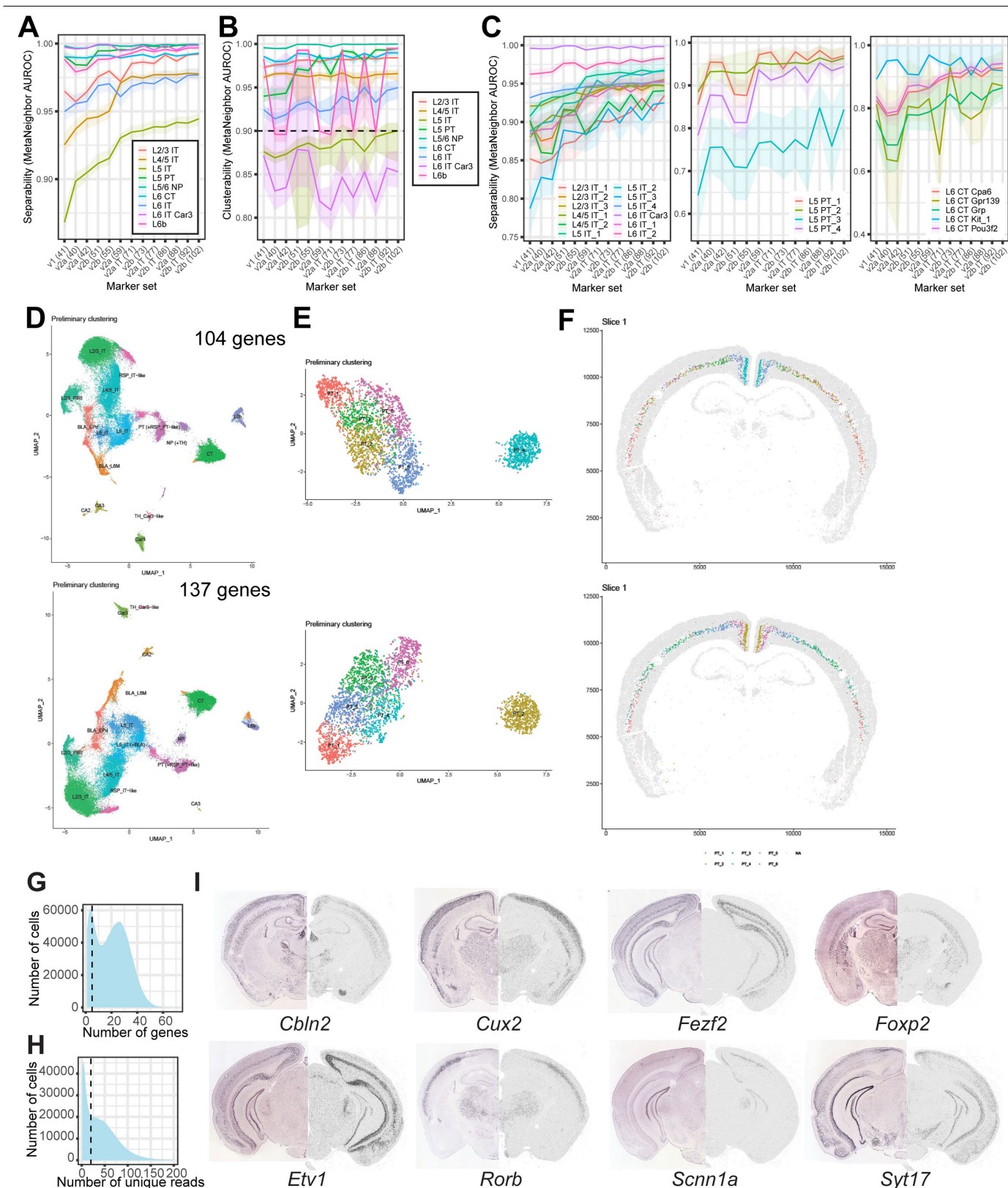

**Extended Data Fig. 1** | See next page for caption.

**Extended Data Fig. 1 | In silico design and evaluation of the gene panel.**
(**A**) In silico assessment of H2 type separability (supervised analysis, ability to distinguish cell types given reference labels) in a surrogate snRNAseq dataset across marker sets ranging from 40 to 102 genes. The number of genes in each panel is shown in parentheses. The final gene panel includes the 102-gene panel, plus two manually added genes (*Fezf2* and *Hgf*). The line shows median performance, the ribbon minimum to maximal performance over 10 independent downsamplings (N = 5000 cells) of the surrogate dataset. (**B**) In silico assessment of H2 type clusterability (unsupervised analysis, ability to recover reference labels through standard clustering) across the marker sets. The line shows median performance, the ribbon first to third quartile over 10 independent downsamplings (N = 5000 cells) of the surrogate dataset. (**C**) In silico assessment of H3 type separability for IT, L5 ET (PT), and L6 CT cells across the marker sets. Line and ribbons defined as in A. (**D**)(**E**) UMAP plots of gene expression of cortical excitatory neurons (D) and L5 ET neurons (E) calculated from the 104-gene panel with or without an additional 33 genes. Colors indicate H2 types in (D) and H3 types in (E). (**F**) Images of a coronal section showing the distribution of L5 ET types clustered using the gene panels. (**G**) Gene counts per cell and (**H**) read counts per cell in the dataset. Quality control thresholds are indicated by dashed lines in both plots. The lower peaks in gene and read counts likely include non-neuronal cells that do not express the cortical neuronal markers in our gene panel and non-cellular particles that are fluorescent. (**I**) The expression patterns of representative genes in Allen Brain Atlas (left half) compared to the current dataset (right half).

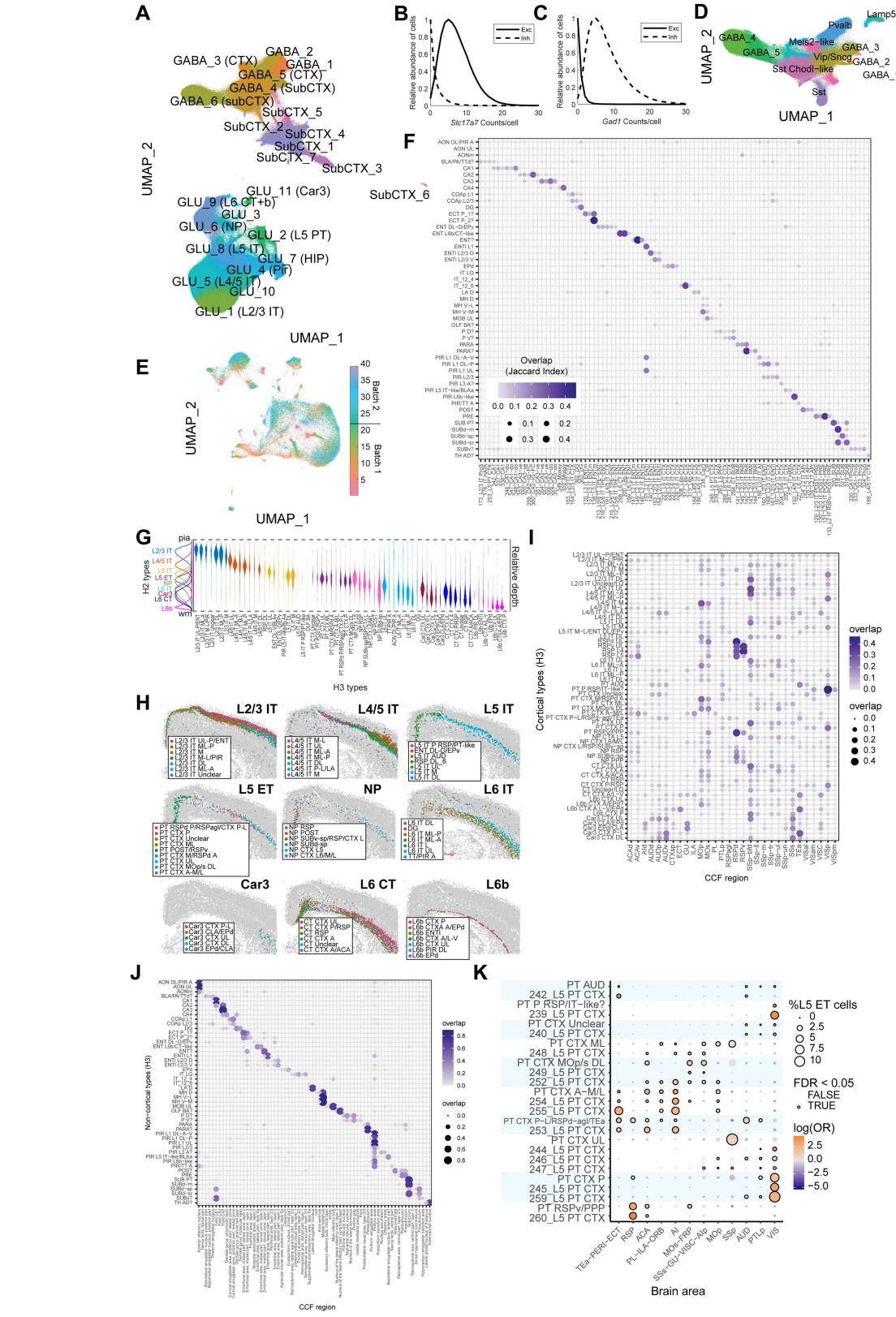

**Extended Data Fig. 2** | See next page for caption.

**Extended Data Fig. 2 | Hierarchical clustering of BARseq data.** (**A**) UMAP plot of the gene expression of all cells. Colors and labels indicate H1 clusters. (**B**)(**C**) Histograms of *Slc17a7* and *Gad1* counts per cell in excitatory and inhibitory neurons. (**D**) UMAP plot of the gene expression of inhibitory neurons. Colors and labels indicate H2 types of inhibitory neurons. (**E**) UMAP plot of the gene expression of excitatory neurons. Colors indicate slice numbers. The coordinates of dots in the UMAP plot are the same as those in Fig. 2c. (**F**) Cluster correspondence between non-isocortical H3 types in BARseq (rows) and single-cell RNAseq (columns)[9]. (**G**) The laminar distribution of H2 types (shown on the left) and each H3 type. H3 types are sorted by their median laminar position. (**H**) The distribution of H3 types in the dorsomedial portion of the cortex on a representative slice. The parent H2 types are indicated in each plot. (**I**)(**J**) Overlap between isocortical (I) and non-isocortical (J) H3 types and CCF-defined areas. (**K**) Distribution across CCF regions of matching L5 ET BARseq H3 types and L5 ET scRNAseq cell types (Jaccard index > 0.1). Matched types are shown next to each other and share the same background color. The colors indicate log odds ratios and circle size indicates the fraction of cells among all L5 ET neurons.

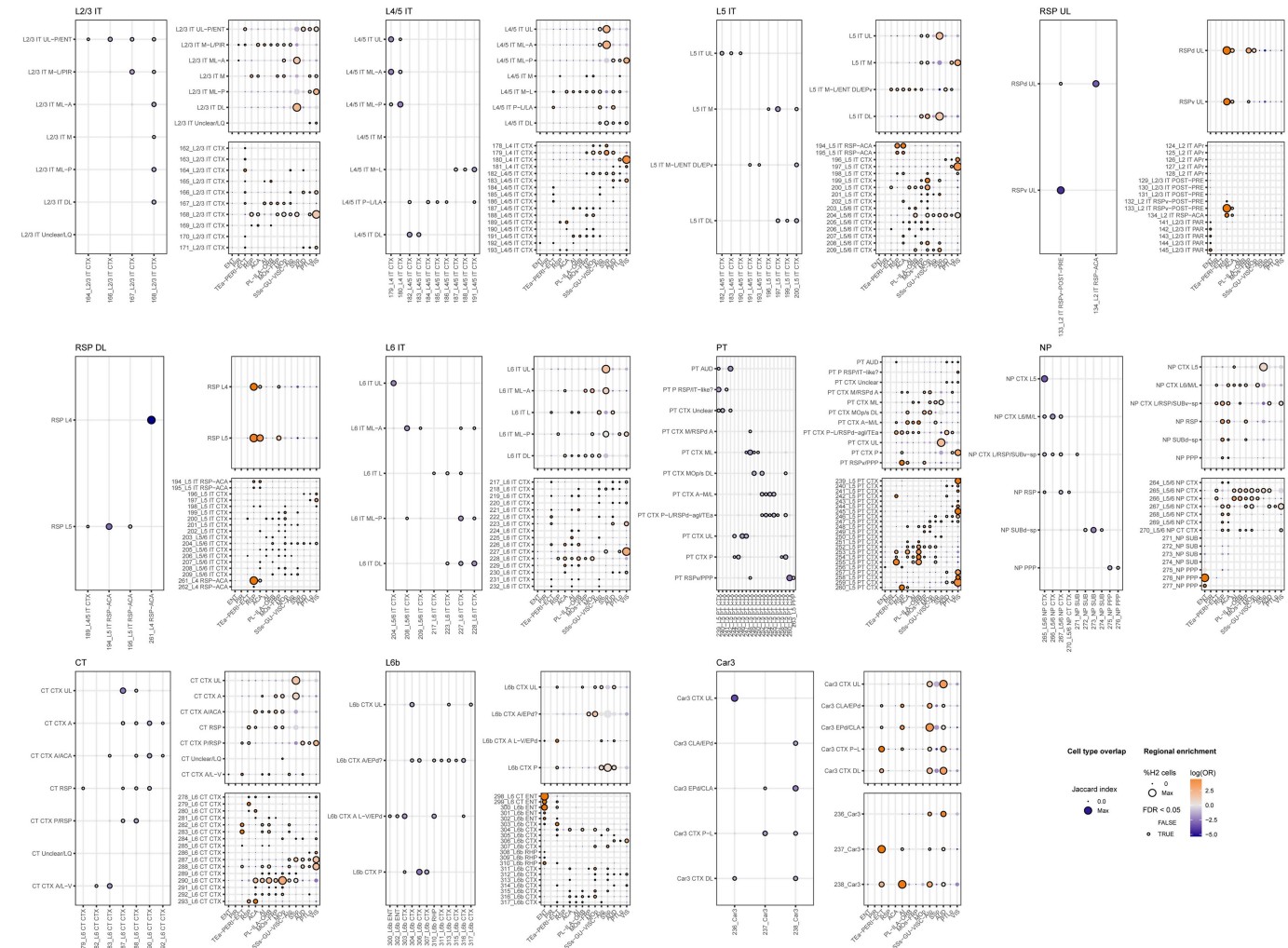

**Extended Data Fig. 3 | Mapping and comparative regional enrichment of BARseq and scRNAseq types.** For each BARseq H2 type, we show the mapping of BARseq H3 types with reference scRNAseq type (*left*), the CCF enrichment of H3 types (*top right*), and the CCF enrichment of scRNAseq types (*bottom left*). The mapping between BARseq and scRNAseq types is quantified as the Jaccard index, significant associations (permutation test) are shown by outlining dots with black circles. The regional enrichment is quantified as odds ratios, significant deviations (hypergeometric test) are shown by outlining dots with

black circles. In the mapping panel, all BARseq H3 types are shown but, for readability, only scRNAseq types with significant associations are plotted. In contrast, the CCF enrichment is shown for all scRNAseq types belonging to subclasses that are equivalent to the BARseq H2 type (e.g., the BARseq L4/5 IT type corresponds to the L4 IT and L4/5 IT subclasses in the scRNAseq dataset). Colors indicate log odds ratios and circle size indicates the fraction of cells among all cells of that H2 type.

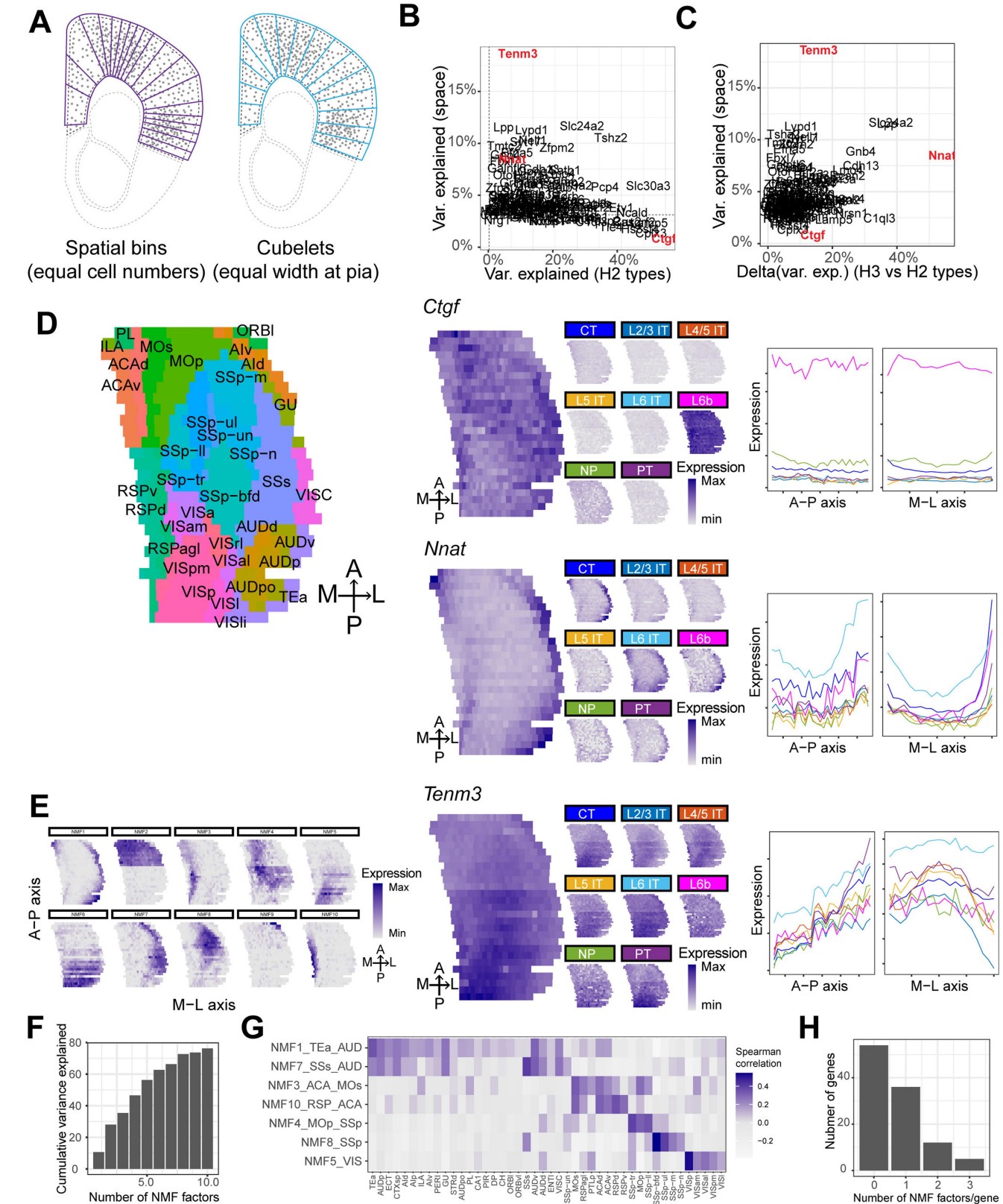

**Extended Data Fig. 4** | See next page for caption.

**Extended Data Fig. 4 | Identifying shared spatial patterns of gene expression using non-negative matrix factorization.** (**A**) Illustrations of the definition of 'spatial bins' used in gene expression analyses (purple outlines) and 'cubelets' used in the analyses of H3 type distributions (blue outlines) in the pilot brain. The definition of spatial bins aimed for equal cell numbers across bins within a slice, whereas the definition of cubelets aimed for equal width on the surface of the cortex. Dots indicate cells. (**B**)(**C**) Variance in gene expression explained by space compared to that explained by H2 types (**B**) or additional variance explained by H3 types (**C**). In (B) dashed lines indicate threshold for p = 0.05 (one-sided F-test with K-1 and 21,197-K degrees of freedom, where K = 8 H2 types or K = 540 spatial bins, Bonferroni corrected for multiple comparisons). (**D**) The expression patterns of the indicated genes (*Ctgf*, *Nnat*, and *Tenm3*) plotted on flatmaps of the cortex in all cells (left column) or in each H2 type (center). The variations of gene expression in each H2 type along the AP axis and the ML axis are shown on the right. Line colors indicate H2 types as shown in the center plots. In the upper left plot, the same map is color-coded and labeled by cortical areas. (**E**) Spatial patterns of all 10 NMF factors. (**F**) Cumulative variance explained by the indicated number of NMF factors. (**G**) Spearman correlation between NMF factors and cortical areas. (**H**) Histogram of the number of NMF factors that each gene is associated with.

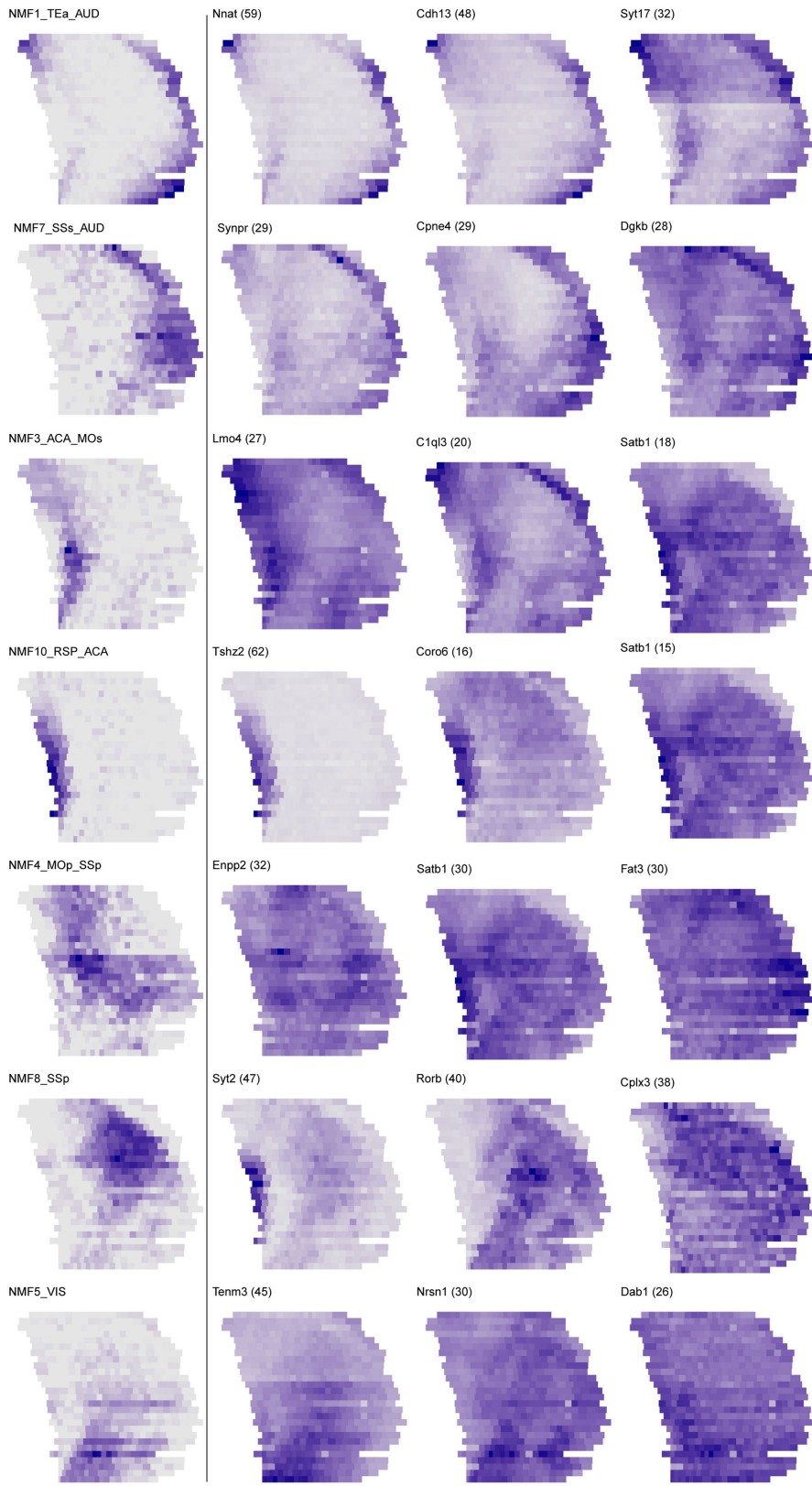

**Extended Data Fig. 5 | Expression patterns of top genes associated with each NMF spatial pattern.** The left column shows the pattern associated with each NMF component; the right column shows the overall expression patterns (total expression counts across all cortical cells) of the top 3 genes associated with each NMF component. Expression patterns were min-max standardized (max expression = blue). Numbers in parentheses next to gene names show the average percentage of gene expression variance explained by the NMF pattern across cortical H2 types.

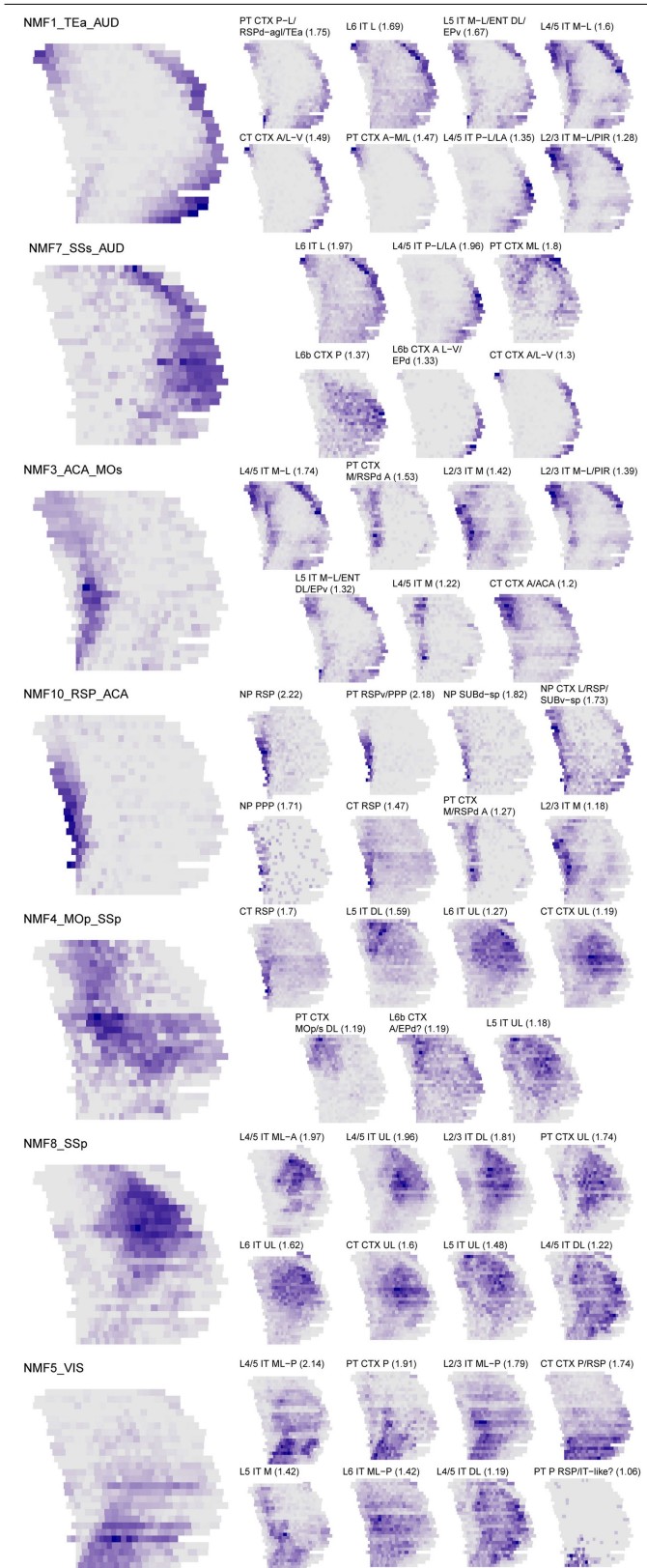

**Extended Data Fig. 6 | Distribution patterns of top H3 types associated with each NMF spatial pattern.** The left column shows the pattern associated with each NMF component; the right column shows the distribution pattern (fraction of cells from the H3 type found in each bin) of H3 types showing above-null association with the NMF pattern. Numbers in parentheses next to H3 type names show the scaled Spearman correlation between the NMF pattern and the distribution pattern.

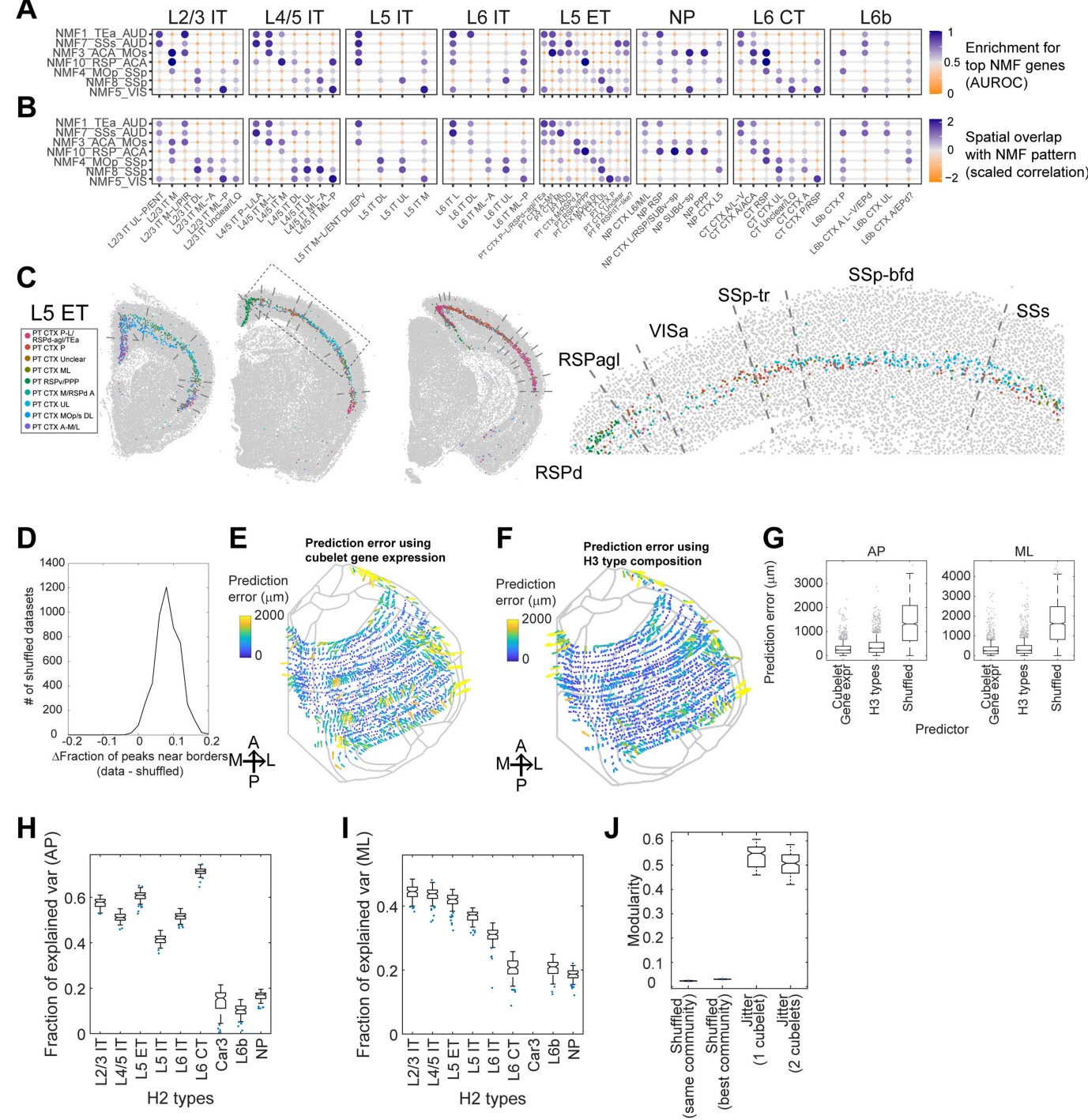

**Extended Data Fig. 7** | See next page for caption.

**Extended Data Fig. 7 | Cortical areas are distinct in H3 type composition.**
(**A**) AUROC of the enrichment for top NMF genes in each H3 type (see
Supplementary Methods). (**B**) Overlap between the spatial patterns of NMF
module expression and the spatial distribution of H3 types. (**C**) The distribution
of L5 ET H3 types in an example coronal section. Dashed lines indicate area
borders in CCF. Magnified views of the dashed boxes are shown on the right.
(**D**) The positions of abrupt changes in the composition of H3 types were shuffled
randomly within each slice, and the difference in the fractions of positions that
were close to a CCF area border between the real data and shuffled data was
calculated (see Supplementary Methods). Positive values indicate that abrupt
changes in the composition of H3 types were more likely to be associated with
area borders in real data than in shuffled control. This shuffling was repeated
5,000 times, and the distribution of this difference is plotted in a histogram.
(**E**)(**F**) Heatmaps showing the errors in predicting cubelet locations using gene
expression (E) or H3 type composition (F). Arrows indicate the directions of
the errors and colors indicate the magnitudes of the prediction errors (in μm).

The lengths of the arrows are proportional to the prediction error. (**G**) Box plots
summarizing the prediction performance shown in (E) and (F). N = 1,651 cubelets
in each column. Boxes show median and quartiles and whiskers indicate range
after excluding outliers. Dots indicate outliers. (**H**)(**I**) The composition of H3
types within each indicated H2 types (x-axes) were used to predict the AP (H)
and ML (I) locations of a cubelet. For each H2 type, we performed n = 100 trials.
In each trial, we randomly held 10% of data as test set to determine the fractions
of variance explained. Boxes show median and quartiles and whiskers indicate
range after excluding outliers. Dots indicate outliers. (**J**) The distribution of
modularity of shuffled data, or data with 1-2 cubelets of jitter in CCF registration
for n = 200 random tests. For shuffled data, we calculated modularity based
on either the same clusters obtained from real data, or by the best clusters
obtained by Louvain community detection on the shuffled data. Boxes show
median and quartiles and whiskers indicate range after excluding outliers. Dots
indicate outliers.

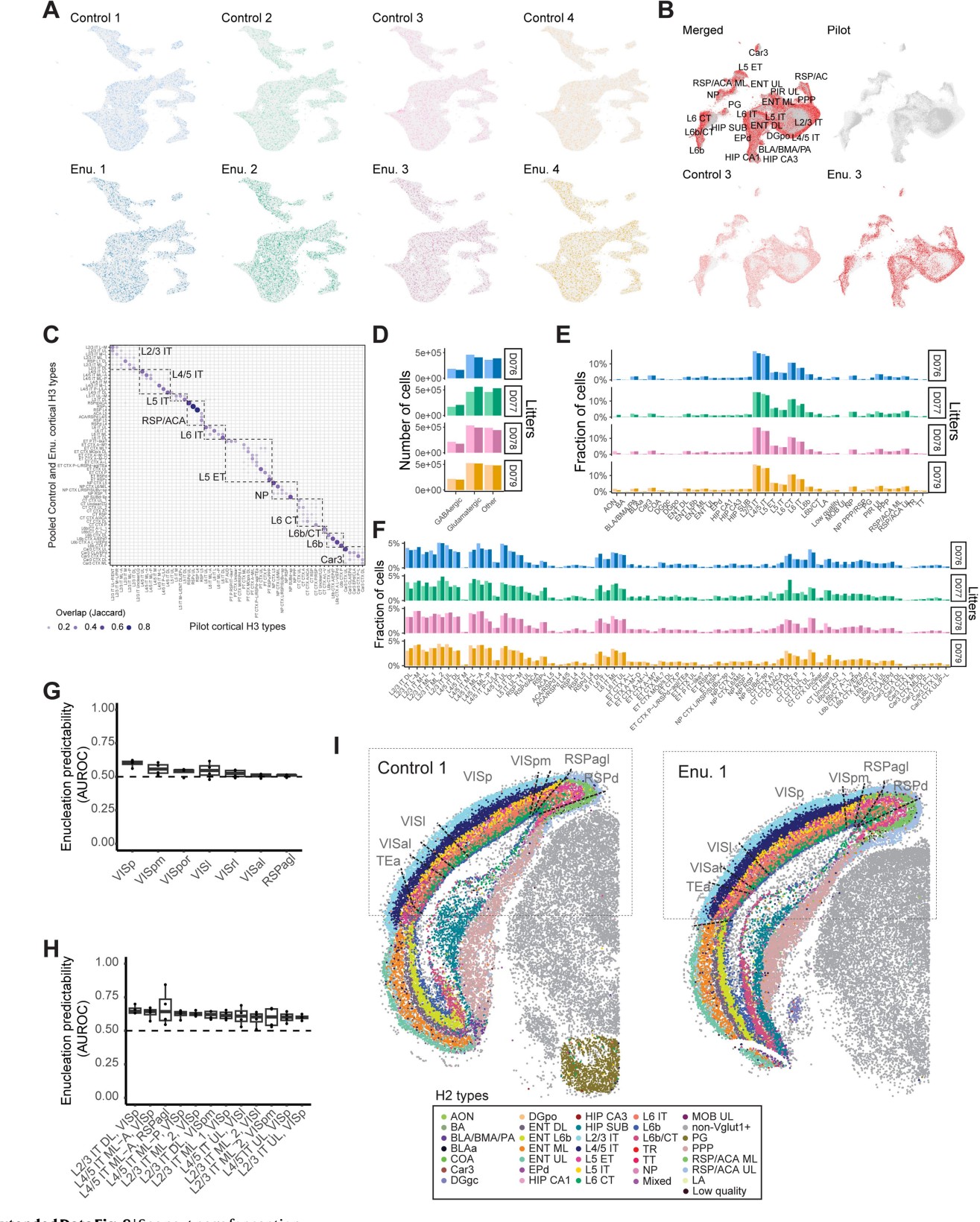

**Extended Data Fig. 8** | See next page for caption.

**Extended Data Fig. 8 | H3 types were consistent between the control brains and the enucleated brains.** (**A**) UMAP plots of the gene expression of neurons from all eight littermate brains. In each plot, neurons from the indicated brain are colored and neurons from all other brains are shown in gray. (**B**) UMAP plot of all excitatory neurons from the pilot brain with excitatory neurons from the eight littermate brains projected onto the same UMAP coordinate space. Merged data on top left shows neurons from the pilot brain in gray and all eight littermate brains in red. Top right shows only neurons from the pilot brain in gray. Bottom row shows excitatory neurons from one pair of littermates. (**C**) Correspondence between H3 types from the eight animals to H3 types in the pilot brain. Dot sizes and colors indicate Jaccard index. Dashed boxes indicate the parent H2 types. (**D**)(**E**) Fractions of cells belonging to each cortical H1 type (D) and H2 type (E) in all paired littermate brains. (**F**) Fractions of cells belonging to each cortical H3 type in all paired littermate brains. In all fraction plots, enucleated animals are represented by the darker color. (**G**) The AUROC scores of a nearest neighbor classifier that predicts the condition (control or enucleated) of a neuron in the indicated cortical areas (N = 4 biologically independent littermate pairs). Only areas with performance over 0.5 for at least 3 out of 4 littermate pairs were shown. Boxes indicate quartiles and medians, and whiskers indicate range. Dots indicate performance for each held-out litter. (**H**) The AUROC scores of a nearest neighbor classifier that predicts the condition (control or enucleated) of a neuron of the indicated H3 types in the indicated cortical areas (N = 4 biologically independent littermate pairs). Only combinations with moderate performance ( > 0.6) were shown. Boxes indicate quartiles and medians, and whiskers indicate range. Dots indicate performance for each held-out litter. (**I**) Same representative slices as shown in Fig. 5e color coded by H2 types. The dashed boxes indicate the area shown in Fig. 5e. Enu, enucleated.

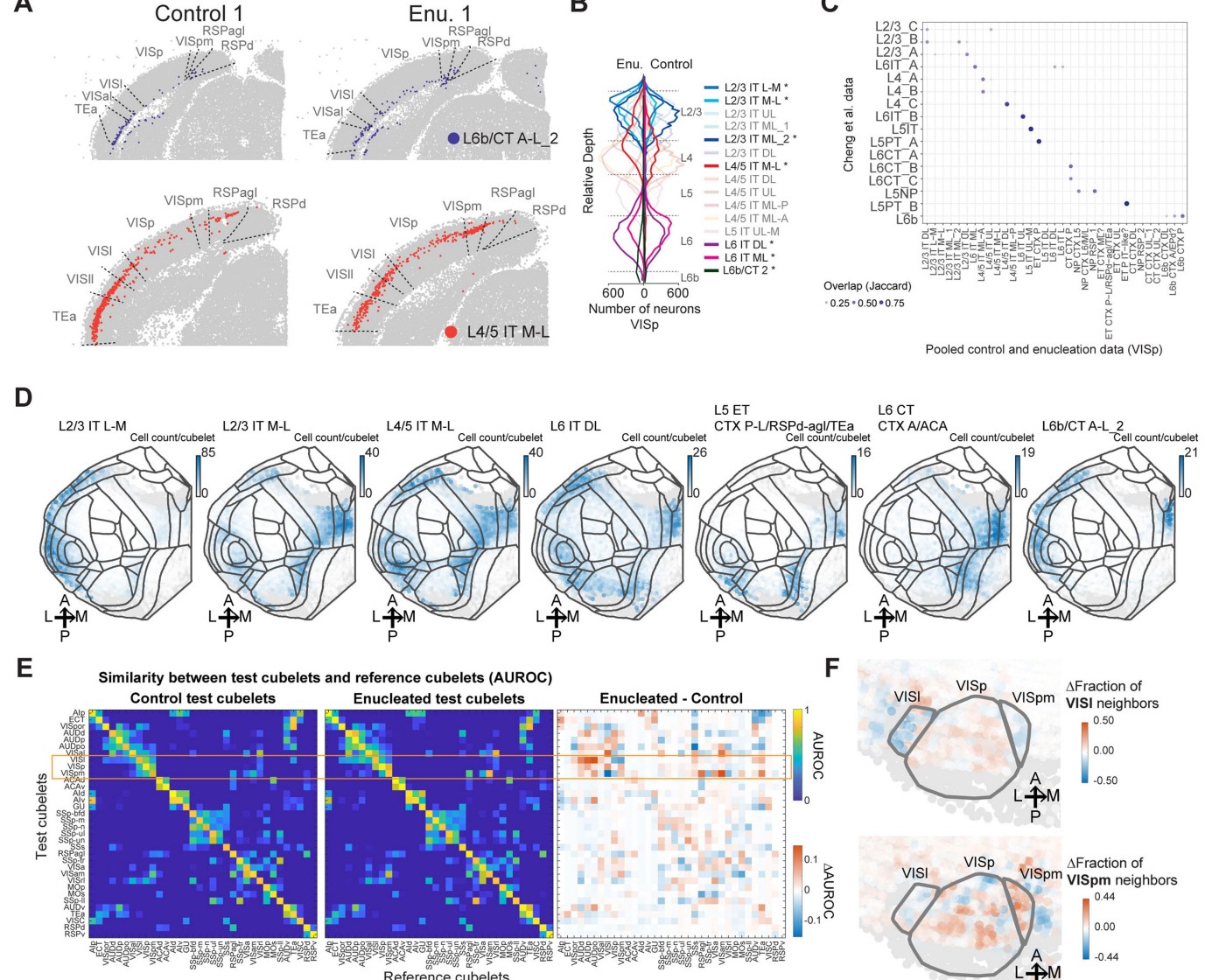

**Extended Data Fig. 9 | Enucleation broadly shifted visual area neurons to H3 types in medial and lateral areas.** (**A**) Example slice images for L6b/CT A-L_2 and L4/5 IT M-L in a representative littermate pair. (**B**) The laminar distribution of H3 types in VISp. H3 types that were enriched or depleted are shown in dark colors. (**C**) Cell type mapping (Jaccard index) of cortical H3 types in the enucleated and control littermates to cell types in Cheng, et al.[42] with or without dark rearing. (**D**) The number of cells per cubelet for the top enriched H3 types in VISp. Colors indicate cell counts in each cubelet. (**E**) AUROC of a nearest neighbor classifier assigning cubelets from control (left) or enucleated (middle) brains to cubelets in reference control brains. The difference in AUROC between the enucleated and the control brains are shown on the right. Orange box highlights the relevant VIS areas (VISp, VISpm, and VISl). (**F**) Magnified views of flatmaps showing the enriched or depleted fraction of VISl (top) or VISpm (bottom) neighbors for each cubelet. The circled areas indicate VISl, VISp, and VISpm. Enu, enucleated.

# Reporting Summary

## Statistics

For all statistical analyses, confirm that the following items are present in the figure legend, table legend, main text, or Methods section.

| n/a | Confirmed | |
|---|---|---|
| ☐ | ☒ | The exact sample size (*n*) for each experimental group/condition, given as a discrete number and unit of measurement |
| ☐ | ☒ | A statement on whether measurements were taken from distinct samples or whether the same sample was measured repeatedly |
| ☐ | ☒ | The statistical test(s) used AND whether they are one- or two-sided *Only common tests should be described solely by name; describe more complex techniques in the Methods section.* |
| ☐ | ☒ | A description of all covariates tested |
| ☐ | ☒ | A description of any assumptions or corrections, such as tests of normality and adjustment for multiple comparisons |
| ☐ | ☒ | A full description of the statistical parameters including central tendency (e.g. means) or other basic estimates (e.g. regression coefficient) AND variation (e.g. standard deviation) or associated estimates of uncertainty (e.g. confidence intervals) |
| ☐ | ☒ | For null hypothesis testing, the test statistic (e.g. *F*, *t*, *r*) with confidence intervals, effect sizes, degrees of freedom and *P* value noted *Give P values as exact values whenever suitable.* |
| ☒ | ☐ | For Bayesian analysis, information on the choice of priors and Markov chain Monte Carlo settings |
| ☒ | ☐ | For hierarchical and complex designs, identification of the appropriate level for tests and full reporting of outcomes |
| ☐ | ☒ | Estimates of effect sizes (e.g. Cohen's *d*, Pearson's *r*), indicating how they were calculated |

*Our web collection on statistics for biologists contains articles on many of the points above.*

## Software and code

Policy information about availability of computer code

| Data collection | Data collection used micro-manager v1.4 and NIS-Elements AR (5.30.04) to drive microscope |
|---|---|
| Data analysis | Custom R (v4.3.0), python(3.8 for cellpose and n2v, 3.9 for bardensr), and MATLAB (2023a) codes were used to process and analyze data. These codes relied on open-source packages, including Bioconductor(v3.18), Cellpose(2.2), Bardensr, n2v(0.3.1), QuickNii(2.2), Visualign(0.9), and FIJI (1.53t). Custom codes are provided on Mendeley Data (https://data.mendeley.com/datasets/8bhhk7c5n9/1 and https://data.mendeley.com/datasets/5xfzcb4kn8/1) and on Github (https://github.com/gillislab/barseq_analysis). |

For manuscripts utilizing custom algorithms or software that are central to the research but not yet described in published literature, software must be made available to editors and reviewers. We strongly encourage code deposition in a community repository (e.g. GitHub). See the Nature Portfolio guidelines for submitting code & software for further information.

## Data

Policy information about availability of data

All manuscripts must include a data availability statement. This statement should provide the following information, where applicable:

- Accession codes, unique identifiers, or web links for publicly available datasets
- A description of any restrictions on data availability
- For clinical datasets or third party data, please ensure that the statement adheres to our policy

Raw sequencing images are available from the Brain Image Library (https://api.brainimagelibrary.org/web/view?bildid=ace-dim-pad, https://

## Research involving human participants, their data, or biological material

Policy information about studies with human participants or human data. See also policy information about sex, gender (identity/presentation), and sexual orientation and race, ethnicity and racism.

| | |
|---|---|
| Reporting on sex and gender | NA |
| Reporting on race, ethnicity, or other socially relevant groupings | NA |
| Population characteristics | NA |
| Recruitment | NA |
| Ethics oversight | NA |

Note that full information on the approval of the study protocol must also be provided in the manuscript.

# Field-specific reporting

Please select the one below that is the best fit for your research. If you are not sure, read the appropriate sections before making your selection.

☒ Life sciences        ☐ Behavioural & social sciences        ☐ Ecological, evolutionary & environmental sciences

For a reference copy of the document with all sections, see nature.com/documents/nr-reporting-summary-flat.pdf

# Life sciences study design

All studies must disclose on these points even when the disclosure is negative.

| | |
|---|---|
| Sample size | Sample sizes were chosen to include duplicates per sex per condition (8 animals), plus an additional brain for pilot study. We saw that the changes associated with enucleation were much stronger than inter-individual variations across replicates/sexes, indicating that our sample size was sufficient. |
| Data exclusions | Cells with low read counts and gene counts were excluded, because they would not be robustly clustered. For part of the analyses, coronal sections from the most anterior and posterior end of the cortex were excluded. These sections were excluded because coronal cuts were not perpendicular to the cortex due to the curvature of the cortex, and so we cannot reliably estimate cell type composition in cubelets drawn on slices. This criteria was pre-established based on the anatomy of the cortex. |
| Replication | The experiments were replicated on four littermate pairs, and the observed effects were consistent across all four replicates. The cell typing results were consistent across all eight littermates and also between the littermates and the pilot brain. All attempted replicates were included in the paper. |
| Randomization | Littermates were randomly assigned to either enucleated or sham condition. |
| Blinding | Because the enucleated animals were easily distinguishable from control animals during both the experiment and data analysis, we did not attempt to blind the experiment during both data collection and analysis. The systematic and comprehensive analysis we performed were not prone to observer biases and thus did not require blinding. This approach conforms with standard practice for -omics studies. |

# Reporting for specific materials, systems and methods

We require information from authors about some types of materials, experimental systems and methods used in many studies. Here, indicate whether each material, system or method listed is relevant to your study. If you are not sure if a list item applies to your research, read the appropriate section before selecting a response.

## Materials & experimental systems

| n/a | Involved in the study |
|---|---|
| ☒ | ☐ Antibodies |
| ☒ | ☐ Eukaryotic cell lines |
| ☒ | ☐ Palaeontology and archaeology |
| ☐ | ☒ Animals and other organisms |
| ☒ | ☐ Clinical data |
| ☒ | ☐ Dual use research of concern |
| ☒ | ☐ Plants |

## Methods

| n/a | Involved in the study |
|---|---|
| ☒ | ☐ ChIP-seq |
| ☒ | ☐ Flow cytometry |
| ☒ | ☐ MRI-based neuroimaging |

## Animals and other research organisms

Policy information about <u>studies involving animals</u>; <u>ARRIVE guidelines</u> recommended for reporting animal research, and <u>Sex and Gender in Research</u>

| Laboratory animals | C57BL/6J male and female 4-8 weeks old mice were used |
|---|---|
| Wild animals | The study did not involve wild animals |
| Reporting on sex | The pilot study involved a male mouse. For experiments in which one littermate each out of four pairs was enucleated, we matched the sex of each littermate pair and analyzed data from four female and four male mice, two enucleated and two control mice of each sex. |
| Field-collected samples | The study did not involve field-collected samples |
| Ethics oversight | All animal procedures were carried out in accordance with the Institutional Animal Care and Use Committee at Cold Spring Harbor Laboratory and John Hopkin's University. |

Note that full information on the approval of the study protocol must also be provided in the manuscript.

## Plants

| Seed stocks | NA |
|---|---|
| Novel plant genotypes | NA |
| Authentication | NA |

