## [Peer Review File · Nature]

Manuscript Title: Whole-cortex in situ sequencing reveals input-dependent area identity

Reviewer Comments & Author Rebuttals

Reviewer Reports on the Initial Version:

Referees' comments:

Referee #1 (Remarks to the Author):

The study by Chen et al generates a transcriptomics-based atlas of excitatory neurons of the mouse cortex by applying BARseq. The authors highlight the high throughput and the low cost compared to other spatial transcriptomics approaches. 40 hemi brain coronal sections have been studied, spatially mapping 1,2 million cells at three levels of granularity, with focus on distinguishing finer transcriptomic excitatory cell types. With this approach they discover a novel organization of neurons in the mouse brain cortex. The quality of the data, analysis and presentations are high, and the conclusions are credible.

While this is extremely impressive, and a few points should be expanded in the manuscript.

- For simplicity, please add a schematic of the padlock probe design to the manuscript.
- How easy or difficult is it to find the threshold to distinguish real signal from background and how were these thresholds selected? How many excitatory, inhibitory and other cells have been removed because of the thresholding?
- How many cells did not express Slc17a7 or Gad1 but could have been classified as excitatory or inhibitory cells, respectively because of the expression of other genes?
- How confident are the authors with the cell segmentation? Please provide images with a larger capture area and higher resolution. It is very difficult to distinguish some of the cells, e.g. the nucleus/cell labeled in cyan could also be multiple cells. Are the DAPI-based outlines expanded?
- How did the authors separate the cortex into the sub-segments?
- The title "Modular cell type organization of cortical areas revealed by in situ sequencing" could be confusing as the focus of this manuscript is on the excitatory neurons. Please rephrase and resent the main advances of the study in the title.

Minor comments:

- Page 4, Line 19: Please add the tissue thickness to the text and a scalebar to Figure 1A.
- Figure 2F/G: A legend for the H3 types is missing to better understand the violin plot and the distribution plot, respectively.
- Figure 5A, B, D: What does the dashed line show?
- Page 3, Line 25. Remove "a" before "neuronal populations".

Referee #2 (Remarks to the Author):

Summary:

The authors present a manuscript that aims to identify if previously identified anatomical and functional cortical modules are represented in the transcriptome. To this end, they adapt and rigorously test a previously published method, BARseq, to allow for low-cost, high-throughput spatial transcriptomics across the brain. The use of multiple existing RNAseq data sets to establish gene panels and validation of their method with published laminar positioning demonstrates a rigorous analysis to unify cortical cell typing across multiple transcriptomic approaches. They then leverage the scalability of BARseq to expand the analysis across multiple cortical brain regions to address the question of how cell types vary across the brain, and contrast those findings to region-invariant cell classes or subclasses. They find that fine grained cell types (here termed H3 types) can appear in more than one brain region; however, brain regions can be identified by the combination of H3 cell types they utilize. Furthermore, anatomical and functional modules previously identified in the literature have very similar combinations of H3 cell types. These results highlight potential transcriptional inroads to manipulate functional modules across the brain and are an exciting expansion in the utility of transcriptomics to capture brain-wide networks. An exploration of how this analysis could lead to new biological discoveries would enhance the impact of this study (as described below). For example, a discussion or demonstration of the developmental origin or biological underpinnings of these transcriptional programs, or a stronger link to their translational potential (as cited in the introduction) would be useful.

Major Points

Scalability & Generalizability:

BARseq expression patterns match very well in the cortex, but the data in Figure 1F would suggest that it is less reliable in other brain areas, such as the dentate gyrus (*Cux2*, *Foxp2*, *Etv1*, *Scnn1a*). Does the high packing density of these cells make cell segmentation and barcode assignment problematic? The authors address this in Supplementary Note 2. Still, such a limitation would make brain-wide analyses more difficult to interpret (the authors cite scalability and low cost as key features of this approach). The authors should include a discussion of this limitation on the upper limit of cell density to reliably segment cells to the main body of the paper. A discussion of any potential issues of optical crowding (and nonlinearity of detection) in small cells when sequencing highly expressed genes (another feature of their gene panel) would also be helpful.

The ability to "generate a high-resolution map of 1.2 million cells with detailed gene expression" is impressive, but the 1.2 million cells include subcortical neurons, which were not analyzed in detail in this paper and may not even be possible with this gene. The authors indicate that the 107 gene panel used is most appropriate for cortical neurons, so the claim of "detailed gene expression" may only be applicable to cortical neurons (517,428—still an impressive number!) and that should be more clear in the introduction.

Panel Genes:

It is unclear how the marker sets in ED Figure 1A-C differ. Supplementary Note 1 and the legend would suggest that genes are added for each panel from left to right. If that is the case, plotting # of genes vs separability would be more informative, especially if there is not a constant increase in the number of genes with each successive gene panel (which is implied here by plotting a continuous line).

The cortex marker gene panel (Table 1) lists only 101 unique genes by my count below, plus the 3 high-level genes. Are there additional genes that bring the total to 107? Also, having this gene list in a more readable format would be useful.

(*Alcam* , *Brinp3* , *C1ql3* , *Calb1* , *Camk4* , *Car3* , *Cbln2* , *Ccn2* , *Cdh13* , *Cdh18* , *Cdh9* , *Col11a1* , *Col19a1* , *Coro6* , *Cplx3* , *Cpne4* , *Cux2* , *Dab1* , *Dcc* , *Dgkb* , *Efna5* , *Enpp2* , *Etv1* , *Fat3* , *Fbxl7* , *Fezf2* , *Foxp2* , *Galnt14* , *Galntl6* , *Gfra1* , *Gnb4* , *Gpc5* , *Gria1* , *Grik1* , *Grin3a* , *Hcn1* , *Hgf* ,

Hs3st2 , Hs3st4 , Hs6st3 , Htr2a , Igsf21 , Il1rapl2 , Inpp4b , Kcnp1 , Kcnn2 , Kctd1 , Lamp5 , Lmo4 , Lpp , Lrrtm4 , Lypd1 , Ncald , Ncam2 , Nell1 , Nnat , Nr4a2 , Nrp1 , Nrsn1 , Nxph1 , Olfm3 , Oprk1 , Otof , Pam , Pcp4 , Prkca , Ptprk , Rab3c , Rasgrf2 , Rasl10a , Rcan2 , Reln , Rora , Rorb , Satb1 , Scnn1a , Sdk1 , Sgcd , Slc24a2 , Slc24a3 , Sorcs3 , Spock3 , Sv2c , Svil , Synpr , Syt17 , Syt2 , Tafa1 , Tafa2 , Tenm3 , Timp2 , Tle4 , Tmsb10 , Tmtc2 , Trhde , Tshz2 , Vwc2l , Zfp804a , Zfp804b , Zfpm2 , Zmat4)

Biological Underpinnings and Disease implications of Modularity

The authors' data overlay a transcriptomic lens on the functional and anatomical modularity of brain circuits. The idea and impact of the manuscript would be strengthened by an exploration or discussion of the potential biological underpinnings that connect transcriptomic, anatomical, and functional networks. A quick gene ontology analysis of the 101 gene panel above shows enrichment for the terms: cell differentiation, cell adhesion, nervous system development, regulation of neuron projection development, and neurogenesis. Do the authors find any genes in the set that are particularly important for distinguishing H3 types rather than classes or subclasses? Furthermore, the authors should comment if any of the 47 RNAseq data sets used to identify the panel included developmental time points. Would exploring this modularity at developmental time points better reveal this modularity? In development, axon guidance molecules and cell adhesion molecules relevant for establishing long-distance wiring to support functional connectivity modules might be most highly expressed. The authors discuss how their data validates prior developmental models (protomat plus extrinsic mechanisms), but it would be helpful to highlight how this data set could lead to new discoveries well.

Do the genes key for distinguishing H3 cell type composition and cortical modularity reveal anything about potential disease states with altered modularity, such as Alzheimer's, MCI, schizophrenia, and depression, as cited in the introduction? For example, Reln is in the panel and has been linked to several of the diseases above. The authors cite a "wire by similarity" rule—could you leverage the spatial power of the authors' approach to identify RELN-enriched modules and their potential role in affecting brain-wide networks identified in human functional connectivity assays?

Minor Points

Figure 1C: There is no information on what "matched sensitivity" means

Figure 2C - The legend indicates that these are excitatory H2 cell types. The text indicates there are 9 H2 cell types (Page 4, Line 4), but this UMAP plot generates 22. Do the authors mean there are 9 cortical H2 cell types and Figure 2C shows all H2 cell types? The rest of the figure and the figure title focus on cortical cell types, so limiting the analysis on B and C to the cortex might clarify this issue.

Figure 2E - RSP suddenly appears as a 10th H2 cell type but is not mentioned anywhere in the text.

Figure 2H - Are both the circle size and the color mapping indicating log(odds ratio)? It is unclear why both would be needed and Figure ED3 uses a different convention. The size and color do not perfectly correlate, so I would assume this is a typo. The values represented should be more clearly indicated in the legend.

Figure 2F. The text claims "H3 types were concentrated in sublayers", but no statistical tests were performed. Sorting by the position of median expression will create an illusion of tiling, so this claim should be explored more quantitatively. Figure 2G. Are colors remapped here to H3 type, rather than H2 type as in Figure 2F? If so, it would be good to indicate it in the legend and with a color bar. These data would also suggest that these differences are perhaps region-specific, such

as in L4/5IT, which is a strength of the BarSeq approach and could be highlighted here. It may also be a reason that the H3 sublayer differences are blurred/underestimated in Figure 2F.

Figure 3D. Labels on the flatmap to orient the reader to A-P and M-L would be helpful.

Figure 4 A-B. Why is H2 type L5ET not included in this list?

Figure 5F. The use of 1-AUROC (as indicated in the methods Page 23, Line 24) to define the heat map is confusing here when the legend refers to a "similarity matrix" and the title states AUROC. Either this labeling should be changed, or the legend should more clearly describe the heat map.

Author Rebuttals to Initial Comments:

While the referees find your work of some interest, they raise concerns about the strength of the novel conclusions that can be drawn at this stage. We feel that these reservations are sufficiently important as to preclude publication of this study in Nature.

In this revision, we added substantial new experiments, including additional BARseq in situ sequencing data of 9.1 million cells from 8 brains, to investigate the roles of peripheral sensory inputs in establishing the characteristic cell type profiles of cortical areas and modules. These new findings address the concerns about the strength of novel conclusions, particularly those from reviewer #2. Using this new dataset, we addressed the following two questions on the development of cortical cell types and areas:

1. Does binocular enucleation generate aberrant cell types? We found that the visual cortex became enriched in neuronal types that were usually found in other areas in a normal brain, but neurons in the visual cortex did not become new cell types.
2. How do the cell type compositional profiles of visual areas change? We found that visual areas shifted towards the compositional profiles of neighboring cortical areas, but only towards areas within the same cortical module.

The new dataset is also a technological feat:

- *An order of magnitude* more data than the original submission: We performed BARseq in *eight additional brains* with or without neonatal binocular enucleation, resulting in *9.1 million additional cells sequenced in situ (total 10.3 million cells with the original data)*.
- High throughput and cost effective: BARseq on all eight animals took 18 days on one microscope, costing ~\$2,000 per brain.
- Highly robust data with little batch effect: High reproducibility of BARseq allowed us to distinguish biological effects associated with enucleation from individual variability across the eight animals.

Thus, in addition to generating new developmental insights, our dataset also provides a proof-of-principle for a new approach: to interrogate and compare brain-wide transcriptomic reorganization across many individuals with different perturbation, genetic background, and/or other conditions. Our study demonstrates that BARseq can go beyond systematic but one-off cell atlasing and *allow brain-wide in situ sequencing to be applied in hypothesis-driven perturbation experiments*. To our knowledge, no other spatial transcriptomic technique has demonstrated the scalability or the reproducibility to do so on a brain-wide scale. We envision that the same strategy can allow other researchers in the broad neuroscience community to address a wide range of questions in neurodevelopment, comparative neurobiology, and neuropsychiatric disorders.

Detailed responses and additional changes are described below.

Referees' comments:

Referee #1 (Remarks to the Author):

The study by Chen et al generates a transcriptomics-based atlas of excitatory neurons of the

mouse cortex by applying BARseq. The authors highlight the high throughput and the low cost compared to other spatial transcriptomics approaches. 40 hemi brain coronal sections have been studied, spatially mapping 1,2 million cells at three levels of granularity, with focus on distinguishing finer transcriptomic excitatory cell types. With this approach they discover a novel organization of neurons in the mouse brain cortex. The quality of the data, analysis and presentations are high, and the conclusions are credible.

While this is extremely impressive, and a few points should be expanded in the manuscript.

We appreciate the reviewer's enthusiasm.

- For simplicity, please add a schematic of the padlock probe design to the manuscript.

We added a schematic of the padlock probe design in Fig. 1A.

- How easy or difficult is it to find the threshold to distinguish real signal from background and how were these thresholds selected? How many excitatory, inhibitory and other cells have been removed because of the thresholding?

The thresholding to distinguish real signals from noise was performed in BarDensr and targets a 5% false detection rate (FDR). The FDR was estimated by counting the number of signals that corresponded to five negative control barcodes (GIs), which were included in the codebook during decoding but not used in padlock probes. This process was fully automated, so no human effort/input was needed. We have clarified this in the Methods in this revision.

The QC at the cell level was separate from the thresholding for colony signals. The QC thresholds for cells were determined by examining the distribution of reads per cell and genes per cell. The original dataset initially contained 2,167,762 cells, and 1,259,256 cells passed QC (58%). We have now added additional description of this process in the Results section.

- How many cells did not express Slc17a7 or Gad1 but could have been classified as excitatory or inhibitory cells, respectively because of the expression of other genes?

Our clustering was based on all genes and did not require Slc17a7 or Gad1 expression as an absolute requirement. Therefore, these clusters already contained cells that were clustered as "excitatory" or "inhibitory," but may not express Slc17a7 or Gad1. Of 427,766 cortical cells in the excitatory neuron clusters, 3,800 did not have detectable Slc17a7 expression. Of 83,394 cortical cells in the inhibitory neuron clusters, 100 did not have detectable Gad1 expression. We have clarified this in the Results in this revision.

- How confident are the authors with the cell segmentation? Please provide images with a larger capture area and higher resolution. It is very difficult to distinguish some of the cells, e.g., the nucleus/cell labeled in cyan could also be multiple cells. Are the DAPI-based outlines expanded?

We have updated Fig. 1B to include a larger and higher resolution image of the segmentation. The segmentation was performed using Cellpose, a machine learning based cell segmentation tool that can use both nuclear and cytoplasmic signals in segmentation. In our experiments, we used DAPI as the nuclear signal and the sequencing signals of all mRNAs as a cytoplasmic

signal. This approach was different from other methods that were based on expanding nuclear signals to find whole cells, and we did not expand our DAPI signals during this process. We have clarified this in the Results in this revision.

Because Cellpose uses both nucleic and cytoplasmic signals, it generally performs very well when the cytoplasmic signals are abundant. Because our gene panel was optimized for cortical excitatory neurons, this generally leads to very good segmentation in the cortex, which can be seen in the larger updated Fig. 1B. In other brain regions, the quality of segmentation would depend on both the density of cells and the expression level of genes in the panel in those cells. Because we focused on the cortex in this study, we did not optimize or assess segmentation in other brain regions.

- How did the authors separate the cortex into the sub-segments?

To cut the cortex into cubelets, we first “un-warped” the cortex on each section into a flat sheet, then cut the un-warped cortex into bins with similar sizes based on cell numbers or width along the pia surface. We have included additional text in the Methods to clarify this process.

- The title “Modular cell type organization of cortical areas revealed by in situ sequencing” could be confusing as the focus of this manuscript is on the excitatory neurons. Please rephrase and present the main advances of the study in the title.

Because the additional experiments constituted a major part of the revised manuscript with exciting new findings, we now changed the title to reflect the new findings on the effect of peripheral inputs on area identities defined by cell type composition.

Minor comments:

- Page 4, Line 19: Please add the tissue thickness to the text and a scalebar to Figure 1A. We have revised Fig. 1A according to the suggestion.

- Figure 2F/G: A legend for the H3 types is missing to better understand the violin plot and the distribution plot, respectively.

We added H3 type names in Fig. 2F/G.

- Figure 5A, B, D: What does the dashed line show?

The dashed lines in the original figures showed the position that corresponded to the most mediodorsal position in the cortex. In the revision, we switched to a visualization that is based on mapping to a flatmap described in Wang et al., 2020 (PMID 32386544) to make this figure visually consistent with the new dataset. In this new visualization, we no longer include the dashed line.

- Page 3, Line 25. Remove "a" before “neuronal populations”.

We have revised this.

Referee #2 (Remarks to the Author):

Summary:

The authors present a manuscript that aims to identify if previously identified anatomical and functional cortical modules are represented in the transcriptome. To this end, they adapt and rigorously test a previously published method, BARseq, to allow for low-cost, high-throughput spatial transcriptomics across the brain. The use of multiple existing RNAseq data sets to establish gene panels and validation of their method with published laminar positioning demonstrates a rigorous analysis to unify cortical cell typing across multiple transcriptomic approaches. They then leverage the scalability of BARseq to expand the analysis across multiple cortical brain regions to address the question of how cell types vary across the brain, and contrast those findings to region-invariant cell classes or subclasses. They find that fine grained cell types (here termed H3 types) can appear in more than one brain region; however, brain regions can be identified by the combination of H3 cell types they utilize. Furthermore, anatomical and functional modules previously identified in the literature have very similar combinations of H3 cell types. These results highlight potential transcriptional inroads to manipulate functional modules across the brain and are an exciting expansion in the utility of transcriptomics to capture brain-wide networks.

We thank the reviewer for his/her enthusiasm.

An exploration of how this analysis could lead to new biological discoveries would enhance the impact of this study (as described below). For example, a discussion or demonstration of the developmental origin or biological underpinnings of these transcriptional programs, or a stronger link to their translational potential (as cited in the introduction) would be useful.

In the revised manuscript, we included a new set of experiments, including whole-hemisphere BARseq data of 9.1 million cells from eight additional animals, which examined the developmental roles of peripheral sensory inputs in the characteristic cell type profiles of cortical areas and modules. Using this new dataset, we addressed the following two questions on the development of cortical cell types and areas:

1. Does binocular enucleation generate aberrant cell types? We found that the visual cortex became enriched in neuronal types that were usually found in other areas in a normal brain, but neurons in the visual cortex did not become new cell types.
2. How do the cell type compositional profiles of visual areas change? We found that visual areas shifted towards the compositional profiles of neighboring cortical areas, but only towards areas within the same cortical module.

These results suggest that the cortical modules, which we defined in the original submission, restrict the possible range of compositional profiles of cortical areas, and peripheral inputs only refine the compositional profiles within that range. *The unprecedented details we observed were enabled by the whole-cortex scale and cellular resolution and allowed us to advance the understanding of how thalamic inputs shape cortical arealization beyond previous studies in the*

field. These new insights immediately suggest many new hypotheses on how thalamic inputs shape cortical arealization, many of which can be tested using the same in situ sequencing-based approach in follow-up studies.

Major Points

Scalability & Generalizability:

BARseq expression patterns match very well in the cortex, but the data in Figure 1F would suggest that it is less reliable in other brain areas, such as the dentate gyrus (*Cux2*, *Foxp2*, *Etv1*, *Scnn1a*). Does the high packing density of these cells make cell segmentation and barcode assignment problematic? The authors address this in Supplementary Note 2. Still, such a limitation would make brain-wide analyses more difficult to interpret (the authors cite scalability and low cost as key features of this approach). The authors should include a discussion of this limitation on the upper limit of cell density to reliably segment cells to the main body of the paper. A discussion of any potential issues of optical crowding (and nonlinearity of detection) in small cells when sequencing highly expressed genes (another feature of their gene panel) would also be helpful.

As with any spatial transcriptomics method, cell segmentation and read assignment are critical to accurately determining cell types in BARseq. Our experiment is highly optimized for the cortex, including in gene panel selection and cell segmentation, and not optimized for other regions. Nonetheless, transcriptomic clusters in cell-dense regions, such as the piriform cortex and the hippocampus still recapitulated cell types in reference single-cell RNAseq datasets. (ED Fig. 2F). Because the raw data of our datasets will be publicly available, the same data can be re-processed for these other regions by other researchers. New segmentation approaches (e.g., Cellpose2, pci-seq, ClusterMap) could potentially improve segmentation especially in these cell-dense regions.

Optical crowding is mostly solved in this dataset by detecting the high expressors in the hybridization cycles. A second strategy that can be used in future experiments is that overlapping colony signals can be demixed computationally (Chen et al., PLoS Comput Biol, 2021, PMID 33684106). This approach was not used in our experiment because we found it to be unnecessary.

Although the current dataset has limitations beyond looking at the cortex, we emphasize that the strength of BARseq lies in its scalability, reproducibility, and generalizability in performing future experiments, potentially targeting different genes for different brain regions and across many individuals. The new experiment of BARseq on eight animals with and without binocular enucleation serves as a proof-of-principle for this approach.

In this revision, we included additional paragraphs in Discussion about potential limitations imposed by segmentation and optical crowding and their solutions, and the general applicability of BARseq to future studies.

The ability to "generate a high-resolution map of 1.2 million cells with detailed gene expression" is impressive, but the 1.2 million cells include subcortical neurons, which were not analyzed in

detail in this paper and may not even be possible with this gene. The authors indicate that the 107 gene panel used is most appropriate for cortical neurons, so the claim of “detailed gene expression” may only be applicable to cortical neurons (517,428—still an impressive number!) and that should be more clear in the introduction.

We have included the number of cortical neurons in the Introduction and the Abstract and clarified that we focus the analyses on the cortex. Although the gene panel was optimized for cortical excitatory neurons, most genes are also expressed in non-cortical neurons and are at least informative of distinguishing high-level neuronal types that are spatially distinct (e.g., Fig. 2B).

Panel Genes:

It is unclear how the marker sets in ED Figure 1A-C differ. Supplementary Note 1 and the legend would suggest that genes are added for each panel from left to right. If that is the case, plotting # of genes vs separability would be more informative, especially if there is not a constant increase in the number of genes with each successive gene panel (which is implied here by plotting a continuous line).

We revised ED Fig. 1A-C to include gene numbers for each panel

The cortex marker gene panel (Table 1) lists only 101 unique genes by my count below, plus the 3 high-level genes. Are there additional genes that bring the total to 107? Also, having this gene list in a more readable format would be useful.

(Alcam , Brinp3 , C1ql3 , Calb1 , Camk4 , Car3 , Cbln2 , Ccn2 , Cdh13 , Cdh18 , Cdh9 , Col11a1 , Col19a1 , Coro6 , Cplx3 , Cpne4 , Cux2 , Dab1 , Dcc , Dgkb , Efna5 , Enpp2 , Etv1 , Fat3 , Fbxl7 , Fezf2 , Foxp2 , Galnt14 , Galntl6 , Gfra1 , Gnb4 , Gpc5 , Gria1 , Grik1 , Grin3a , Hcn1 , Hgf , Hs3st2 , Hs3st4 , Hs6st3 , Htr2a , Igsf21 , Il1rapl2 , Inpp4b , Kcnip1 , Kcnn2 , Kctd1 , Lamp5 , Lmo4 , Lpp , Lrrtm4 , Lypd1 , Ncald , Ncam2 , Nell1 , Nnat , Nr4a2 , Nrp1 , Nrsn1 , Nxph1 , Olfm3 , Oprk1 , Otof , Pam , Pcp4 , Prkca , Ptprk , Rab3c , Rasgrf2 , Rasl10a , Rcan2 , Reln , Rora , Rorb , Satb1 , Scnn1a , Sdk1 , Sgcd , Slc24a2 , Slc24a3 , Sorcs3 , Spock3 , Sv2c , Svil , Synpr , Syt17 , Syt2 , Tafa1 , Tafa2 , Tenm3 , Timp2 , Tle4 , Tmsb10 , Tmtc2 , Trhde , Tshz2 , Vwc2l , Zfp804a , Zfp804b , Zfpm2 , Zmat4)

The reviewer is correct that we only had 101 genes plus 3 high-level markers in the panel (total 104 genes). We originally designed probes for more genes. Several of them did not yield padlock probes that fit our design criteria, and we miscounted which genes those were. We corrected this mistake and included a new spreadsheet in Supplementary Table 1 that list all genes and which cell types they distinguish, as the reviewer suggested.

Biological Underpinnings and Disease implications of Modularity

The authors' data overlay a transcriptomic lens on the functional and anatomical modularity of brain circuits. The idea and impact of the manuscript would be strengthened by an exploration or discussion of the potential biological underpinnings that connect transcriptomic, anatomical, and functional networks. A quick gene ontology analysis of the 101 gene panel above shows enrichment for the terms: cell differentiation, cell adhesion, nervous system development, regulation of neuron projection development, and neurogenesis. Do the authors find any genes in the set that are particularly important for distinguishing H3 types rather than classes or subclasses?

Because gene panels only include a subset of all genes that best distinguish cell types, we are reluctant to draw new conclusions based on the specific set of marker genes we chose. Instead, in this revision, we chose to focus on the developmental processes that produced the cell type organization across cortical areas described in the original submission (see below for details).

Furthermore, the authors should comment if any of the 47 RNAseq data sets used to identify the panel included developmental time points. Would exploring this modularity at developmental time points better reveal this modularity? In development, axon guidance molecules and cell adhesion molecules relevant for establishing long-distance wiring to support functional connectivity modules might be most highly expressed. The authors discuss how their data validates prior developmental models (protomap plus extrinsic mechanisms), but it would be helpful to highlight how this data set could lead to new discoveries well.

In the revised manuscript, we included an additional set of experiments that examined the roles of peripheral sensory inputs from the thalamus (i.e., extrinsic mechanisms in prior developmental models) in establishing the characteristic cell type profiles of cortical areas and modules. We interrogated the same cell type markers in whole cortical hemispheres across eight additional animals, with or without binocular enucleation (total 9.1 million cells). We found that:

- (1) Enucleation did not generate new cell types, but neurons in the visual cortex became enriched in cell types that were commonly found in other cortical areas in a normal brain.
- (2) Primary and secondary visual areas became more similar to nearby cortical areas that were further away from the primary visual cortex and within the visio-auditory cortical module.

These results suggest that peripheral sensory inputs sharpen the distinction of the cell type compositional profiles mainly within cortical modules. We further discussed implications of these results in the context of cortical development in the Discussion of this revised manuscript.

The 47 RNAseq datasets were all adult data. In this revision, we clarified this in Supplementary Note 1.

Do the genes key for distinguishing H3 cell type composition and cortical modularity reveal anything about potential disease states with altered modularity, such as Alzheimer's, MCI, schizophrenia, and depression, as cited in the introduction? For example, Reln is in the panel and has been linked to several of the diseases above. The authors cite a "wire by similarity" rule—could you leverage the spatial power of the authors' approach to identify RELN-enriched modules and their potential role in affecting brain-wide networks identified in human functional connectivity assays?

Because genes that distinguish H3 types are likely not unique and the genes that we picked were not necessarily relevant to the specification or functions of cell types, we are hesitant to draw conclusions on the disease implications of individual genes that are distinguishable across H3 types. However, as discussed above, the new experiments demonstrate an even stronger approach: to establish causal relationships between developmental perturbations and cell type

organization. In the Discussion, we included a new paragraph on how a similar approach can be broadly applied to other studies, including brain-wide cell type changes in disease models.

Minor Points

Figure 1C: There is no information on what “matched sensitivity” means. Matched sensitivity means we subsampled single-cell RNAseq datasets to mimic the relatively lower sensitivity of BARseq compared to single-cell approaches. We have clarified this in the legends of Fig. 1C in the revision.

Figure 2C - The legend indicates that these are excitatory H2 cell types. The text indicates there are 9 H2 cell types (Page 4, Line 4), but this UMAP plot generates 22. Do the authors mean there are 9 cortical H2 cell types and Figure 2C shows all H2 cell types? The rest of the figure and the figure title focus on cortical cell types, so limiting the analysis on B and C to the cortex might clarify this issue.

The reviewer is correct that there are 9 H2 types that are shared across the cortex, whereas Fig. 2C shows all H2 excitatory types. In this revision, we clarified in Fig. 2C legend that this included all H2 types.

Figure 2E - RSP suddenly appears as a 10th H2 cell type but is not mentioned anywhere in the text.

In the revision, we clarified that there are 9 H2 types that are shared across all cortical areas, and one H2 type (RSP) that is specific to the medial cortex.

Figure 2H - Are both the circle size and the color mapping indicating log(odds ratio)? It is unclear why both would be needed and Figure ED3 uses a different convention. The size and color do not perfectly correlate, so I would assume this is a typo. The values represented should be more clearly indicated in the legend.

We changed ED Fig. 3 and Fig. 2H so that their styles are consistent. We further clarified what colors and dot-size indicate in the legends in these figures.

Figure 2F. The text claims “H3 types were concentrated in sublayers”, but no statistical tests were performed. Sorting by the position of median expression will create an illusion of tiling, so this claim should be explored more quantitatively.

With Fig. 2F, we only wanted to highlight that each H3 type is usually more confined in layers than the parent H2 type, which is consistent with previous studies. The laminar distribution of H3 types was not used in subsequent analyses. As the reviewer suggested, we performed one-way ANOVA to test whether H3 types within an H2 type indeed were localized to distinct locations. We have added this to this section of the Result.

Figure 2G. Are colors remapped here to H3 type, rather than H2 type as in Figure 2F? If so, it would be good to indicate it in the legend and with a color bar. These data would also suggest that these differences are perhaps region-specific, such as in L4/5IT, which is a strength of the BarSeq approach and could be highlighted here. It may also be a reason that the H3 sublayer differences are blurred/underestimated in Figure 2F.

We agree with the reviewer that the H3 type sublayer differences are blurred in Fig. 2F partially because of area-specific distribution, especially since different cortical areas have different depth for each layer. Because the rest of the manuscript (Fig. 3 –Fig. 8) focus on area localization only and not laminar positions, we did not try to tease apart these two factors.

We have changed the color family used in Fig. 2G to avoid confusion and added cell type legends in both 2F and 2G.

Figure 3D. Labels on the flatmap to orient the reader to A-P and M-L would be helpful. We have added AP/ML axes for Fig. 3D, and a reference map to show where each cortical area is.

Figure 4 A-B. Why is H2 type L5ET not included in this list?

In this figure “L5 ET” is accidentally labeled as “PT,” which was a commonly used name for the same cell type in previous literature. We have changed this back to L5 ET.

Figure 5F. The use of 1-AUROC (as indicated in the methods Page 23, Line 24) to define the heat map is confusing here when the legend refers to a “similarity matrix” and the title states AUROC. Either this labeling should be changed, or the legend should more clearly describe the heat map.

The plotted values are the AUROC for classification between a pair of areas (i.e., a pair of dissimilar areas would have a score of 1). We have corrected the description in the Methods and removed the mentioning of “similarity matrix” in the legend.

Reviewer Reports on the First Revision:

Referees' comments:

Referee #1 (Remarks to the Author):

All the points I raised have satisfactorily been addressed in the revised version. The new added data has also improved the manuscript considerably.

Referee #2 (Remarks to the Author):

The authors present a revised manuscript that rigorously quantifies the spatial distribution of cortical cell types in the mouse. This version significantly expands the impact of their work, both in the technical and conceptual domains. In the technical domain, the authors further demonstrate that they can adapt BARseq for robust, reproducible, and scalable spatial transcriptomics across many animals and cells. The ability of their approach to avoid batch effects, as well as the scalability to multiple animals, will further enhance the potential of BARseq for spatial transcriptomics. These experiments include over 1 million cells across multiple animals—a truly impressive number. The scale of this data set allows for comparison across treatment conditions or disease states, as they nicely demonstrate in the additional experiments added in review.

The authors leveraged this scalability to probe additional questions in the conceptual domain. Sensory perturbations during developmental critical periods have long been known to trigger spatial reorganization of cortical maps. Diminished input can cause a contraction of cortical territory and an expansion of the surrounding territory, traditionally assessed by receptive field mapping, but more recent work has also highlighted the transcriptional changes that occur during sensory deprivation. The authors use the capabilities of BARseq and H3 cell typing to investigate changes in cell type composition after removing visual input via enucleation. Prior work using single-cell RNAseq has also demonstrated that visual deprivation can trigger cell type changes, particularly in Layer 2/3 (Cheng et al 2022). As a result, the exact biological findings here are not unexpected, but they provide much better spatial resolution than the scRNA seq, allowing the authors to make more concrete conclusions about the cell type composition of areas at the interface between cortical areas with spared/deprived inputs. Moreover, this dataset nicely demonstrates the capabilities of BARseq's scalability.

Taken together, the authors provide an impressive data set, a rigorous analysis, and have gone above and beyond to address my initial comments and concerns on the manuscript. I would enthusiastically support this paper for publication in Nature.